# Satellite Observations of Snowfall Regimes over the Greenland Ice Sheet

Elin A. McIlhattan[1], Claire Pettersen[2], Norman B. Wood[2], and Tristan S. L'Ecuyer[1]

[1]Department of Atmospheric and Oceanic Sciences, University of Wisconsin-Madison, Madison, Wisconsin, USA
[2]Space Science and Engineering Center, University of Wisconsin-Madison, Madison, Wisconsin, USA

**Correspondence:** Elin A. McIlhattan (mcilhattan@wisc.edu)

**Abstract.** The mass of the Greenland Ice Sheet (GrIS) is decreasing due to increasing surface melt and ice dynamics. Snowfall both adds mass to the GrIS and has the capacity to reduce surface melt by increasing surface brightness, reflecting additional solar radiation back to space. Modeling the GrIS's current and future mass balance and potential contribution to future sea level rise requires reliable observational benchmarks for current snowfall accumulation as well as robust connections between individual snowfall events and the large-scale atmospheric circulation patterns that produce them. Previous work using ground-based observations showed that, for one research station on the GrIS, two distinct snowfall regimes exist: those associated with exclusively ice-phase cloud processes (IC) and those involving mixed-phase processes indicated by the presence of super-cooled liquid water (CLW). The two regimes have markedly different accumulation characteristics and dynamical drivers. This study leverages the synergy between two satellite instruments, CloudSat's Cloud Profiling Radar (CPR) and CALIPSO's Cloud-Aerosol Lidar with Orthogonal Polarization (CALIOP), to identify snowfall cases over the full GrIS and partition them into the IC and CLW regimes. We find that overall, most CPR observations of snowfall over the GrIS come from IC events (70 %), however, during the summer months, close to half of the snow observed is produced in CLW events (45 %). IC snowfall plays a dominant role in adding mass to the GrIS, producing $\sim$80 % of the total estimated 399 Gt yr$^{-1}$ accumulation. Beyond the cloud phase that defines the snowfall regimes, the macrophysical cloud characteristics are distinct as well; the mean IC geometric cloud depth ($\sim$4 km) is deeper than the CLW geometric cloud depth ($\sim$2 km), consistent with previous studies based on surface observations. Two-dimensional histograms of the vertical distribution of CPR reflectivities show that IC events demonstrate consistently increasing reflectivity toward the surface while CLW events do not. Analysis of ERA5 reanalyses shows that IC events are associated with cyclone activity and CLW events generally occur under large scale anomalously high geopotential heights over the GrIS. When combined with future climate predictions, this snapshot of GrIS snowfall characteristics may shed light on how this source of ice sheet mass might respond to changing synoptic patterns in a warming climate.

## 1 Introduction

There is enough freshwater stored in the Greenland Ice Sheet (GrIS) to raise sea levels globally by 7.36 m (Bamber et al., 2013). Up until the 1990s, generally the mass gained from precipitation balanced the mass lost from melt runoff and ice dynamics in the margins (Zwally et al., 2011; van den Broeke et al., 2016; Mouginot et al., 2019). However, in recent decades the GrIS

has been consistently losing mass (e.g. Zwally et al., 2011; van den Broeke et al., 2016; Mouginot et al., 2019; Mottram et al., 2019). Between 1972 and 2018, the GrIS contributed 13.7 mm to global sea level rise (Mouginot et al., 2019) and by the end of the century the ice sheet is predicted to contribute up to 15 cm to the global mean sea level (Vaughan et al., 2013).

Snowfall is responsible for both adding mass to and brightening the surface of the GrIS. While vapor deposition can be locally important, snowfall accumulation is by far the largest source term for the mass of the GrIS (Ettema et al., 2009; Bring et al., 2016). Snowfall rate, duration, and frequency of events are all important for accumulation. Surface brightness, or albedo, is largely dependent on the frequency of precipitation because fresh snow is more reflective than old snow and bare ice in the shortwave solar wavelengths (Petty, 2006; Box et al., 2012; Ryan et al., 2019). However, the shortwave albedo only matters during sunlit periods when there is incoming solar radiation; therefore the seasonal timing of snowfall events must also be considered. Fresh snowfall in summer can reduce absorbed shortwave by up to a factor of ∼3, largely reducing local melt and meltwater runoff (Noël et al., 2015).

Snowfall characteristics depend on atmospheric conditions, regional surface properties, and topography. There is consensus among modeling studies and observational datasets that most of the GrIS snowfall is produced by cyclones, with the highest accumulation occurring where their moist air masses move up the steep orography of the southeastern coastline (e.g. Kapsner et al., 1995; Schuenemann et al., 2009; Hakuba et al., 2012; Vihma et al., 2016; Berdahl et al., 2018). One encounters less agreement when it comes to the total amount of snowfall over the full GrIS. Ground-based observations provide detailed snowfall information but are subject to spatial and/or temporal limitations. Automated weather stations can indirectly measure snowfall using changes in surface height, providing high temporal resolution data that can resolve accumulation from individual storms (Steffen and Box, 2001), however the observations are limited to the particular location of the stations and their time period of operation. Both airborne and ground-based radars have been used to detect internal reflecting horizons below the surface to provide historical accumulations over the GrIS (Miège et al., 2013; Lewis et al., 2017), but those values are limited to specific transects, suffer complications from melt events, and only apply on annual or longer timescales. Estimating snowfall frequency and accumulation over the whole ice sheet is often achieved using regional climate models (e.g. Berdahl et al., 2018; Mouginot et al., 2019) or reanalyses (e.g. Schuenemann et al., 2009). There is evidence that reanalyses greatly overestimate southern coastal (Bromwich et al., 2016; Ryan et al., 2020) and inland GrIS snowfall (Koyama and Stroeve, 2019) relative to available surface observations. Bromwich et al. (2016) found the ERA-Interim reanalysis has an annual mean precipitation bias of ∼50 % for two stations on the southern coastline of Greenland, while Koyama and Stroeve (2019) found that the Arctic System Reanalysis has snowfall rates more than double that of the surface observations. Even with identical forcing, regional models have been shown to produce a wide range of GrIS precipitation amounts (Vernon et al., 2013), further highlighting the need for better understanding of the connections between large-scale atmospheric conditions and snowfall. In order to predict how snowfall on the ice sheet may change in the future, it is not sufficient to merely know how much snow accumulates in the present but also understand the processes and large-scale drivers that produce it.

Given the scarcity of ground-based snowfall observations, satellites are useful tools for detecting and quantifying snowfall across the GrIS. Surface snowfall can be estimated from space based on the upwelling microwave emission from the surface being diminished by scattering due to ice and snow particles in the atmosphere (Liu and Curry, 1997). However, such satellite

retrievals must assume a priori the upwelling surface emission, and since microwave emissivity varies substantially depending on surface conditions (fresh snow, old snow, ice, meltwater content) large uncertainties are introduced when the retrieval is applied over non-ocean surfaces. Passive microwave sensors can also provide information on the extent, and in some conditions depth, of the snowpack (Frei et al., 2012), however, they measure snow already on the ground which can be impacted by processes other than snowfall (e.g. blowing snow, melt events, etc) and do not give information about the clouds that produce the snow. Satellite-borne active sensors are an advantageous platform for measuring the annual cycle of snowfall over the full GrIS because they can provide both information on falling snow as well as insight into the coincident clouds. In recent years, the Cloud Profiling Radar (CPR) aboard NASA's CloudSat satellite has provided unprecedented insight into snowfall processes in remote, ice-covered regions (e.g. Palerme et al., 2014; Norin et al., 2015; Palerme et al., 2016; Milani et al., 2018; Souverijns et al., 2018; Palerme et al., 2019). Two recent studies have used CloudSat's CPR to look at snowfall over the GrIS in particular: Lenaerts et al. (2020) focused on GrIS snowfall frequency and leveraged the satellite observations to evaluate climate model output; and Bennartz et al. (2019) used the radar measurements to provide the first in-depth, observationally based snowfall rate estimates of the GrIS.

Using detailed, ground-based measurements at Summit Station, a research facility in the center of the GrIS, Pettersen et al. (2018) (hereafter P18) showed that there are distinct atmospheric processes associated with snowfall events that originate from either ice clouds or from Arctic mixed-phase clouds. P18 used microwave radiometers (MWRs) to partition snowfall events into the two regimes: those produced by fully-glaciated ice clouds and those produced by Arctic mixed-phase clouds containing super-cooled liquid water (hereafter IC and CLW events, respectively). P18 highlighted that each precipitation regime exhibited marked differences in cloud microphysical properties, associated atmospheric circulations, and air mass origins.

P18 found that IC events at Summit are associated with deep clouds that advect moist air quickly up and over the southeast Greenland coastline and on to the central GrIS. The North Atlantic cyclones that set up these cloud systems have been credited in many studies for producing snowfall over Greenland (e.g. Serreze and Barrett, 2008; Schuenemann et al., 2009; Berdahl et al., 2018). Conversely, P18 found that CLW events are associated with shallow clouds and slow-moving, quiescent air masses originating from the south and southwest coastlines. The high surface pressure anomaly associated with these conditions was shown by Hanna et al. (2016) to have mainly positive precipitation anomalies over the GrIS in reanalyses. By dividing Summit snowfall by cloud phase, P18 illustrated that IC and CLW events at that particular location have distinct large-scale dynamical drivers which may respond differently to the rapidly changing Arctic climate.

In this study, we aim to expand the ground-based snowfall regime analysis of P18 to the full GrIS, exploring the importance of cloud phase to both snowfall frequency and accumulation. Flying in the same NASA satellite constellation as CloudSat, the Cloud–Aerosol Lidar and Infrared Pathfinder Satellite Observations (CALIPSO) satellite carries the Cloud-Aerosol Lidar with Orthogonal Polarization (CALIOP) instrument which is highly sensitive to the cloud liquid layer at the top of Arctic mixed-phase clouds and can thus reliably discern cloud phase (Matus and L'Ecuyer, 2017; McIlhattan et al., 2017; Morrison et al., 2018). In this work, we use cloud phase data from CALIOP to divide snowfall events identified by CloudSat's CPR into IC and CLW regimes. Leveraging the synergy of the two instruments, we:

– Quantify the percentage of IC and CLW events that are likely missed by CloudSat using ground-based instrumentation

- Map the seasonal frequency of snowfall over the GrIS and show the relative contributions from IC and CLW events

- Quantify the total accumulation of snowfall and the fraction resulting from each regime

- Separate snowfall observations by elevation to see how frequency and rate vary with height for the regimes

- Compare the satellite observations to ground-based data from Summit Station

- Examine the annual cycles of precipitation regimes and discuss their importance for adding mass and brightening the GrIS

- Document the average cloud properties of the two precipitation regimes and their full distributions

- Map the atmospheric circulations that favor each regime in different regions: near-Summit, southeastern, western, and northern GrIS

In the following section we describe the datasets and methods used in this study (Section 2). In Section 3, we compare CloudSat's CPR to surface observations, showing that the CPR is capable of detecting ∼95 % of IC events and ∼75 % of CLW events. We go on to examine the distinctions between IC and CLW events in Section 4. Looking first at snowfall frequency and accumulation (Section 4.1), we find that the IC snowfall events are overall more frequent and have higher snowfall rates than the CLW events. IC snowfall therefore plays the dominant role in adding mass to the GrIS, producing ∼80 % of the total annual accumulation. However, we find the CLW events to be nearly as frequent as IC events in the summer months, meaning that CLW events play an important role in brightening the GrIS during the time of greatest solar insolation. We go on to examine the differences in clouds characteristics for two regimes (Section 4.2). Clouds associated with IC snowfall are consistently both geometrically deeper and have larger integrated reflectivity values than clouds associated with CLW snowfall. The final distinction we look at is in atmospheric circulation patterns coincident with regime snowfall in four regions of the GrIS (Section 4.3) We find evidence that varied cyclone locations are associated with IC events in each region, while CLW events in all regions occur under anomalously high pressure scenarios. We summarize our conclusions in Section 5.

## 2   Datasets and Methods

To explore the connection between cloud phase and the rate and frequency of snowfall over the Greenland Ice Sheet (GrIS), we use space-borne observations from NASA's A-Train satellite constellation. We employ a product developed using instruments at Summit Station as well as ground-based radar measurements to independently corroborate the regime behavior observed by the satellites. We leverage reanalysis output to investigate what large-scale atmospheric patterns are coincident with the two precipitation regimes.

Throughout our analysis, we divide our data into two seasons: summer (May - September) and winter (October - April). This is consistent with the seasonal breakdown of P18 which was based on the distinct atmospheric conditions that occur at Summit during those time periods. Summer conditions in the central GrIS (increased snowfall frequency, higher precipitable water

vapor, more frequent liquid containing clouds, warmer temperatures, lower wind speeds, etc.) all persist through September (Shupe et al., 2013; Castellani et al., 2015). This summer/winter distinction extends outside of the central GrIS as well. Zhang et al. (2004) used a similar two season definition (winter: October – March, summer: April - September), finding that winter has greater cyclone intensity in the North Atlantic and more storms close to the SE coast of Greenland relative to the summer.

## 2.1 Satellite Data

NASA's A-Train satellites orbit at a height of 705 km and a 98.2° inclination in a sun-synchronous orbit, providing detailed observations of the atmosphere and underlying terrain from 82° S to 82° N (L'Ecuyer and Jiang, 2010). The CloudSat and CALIPSO satellites joined the A-Train in 2006 and their close positioning has allowed for over 10 years of collocated observations of the vertical distribution of clouds and precipitation. The 94-GHz CPR aboard CloudSat has a minimum detectable reflectivity factor of -30 dBZe and is sensitive to large cloud particles and hydrometeors (Tanelli et al., 2008). CALIPSO carries CALIOP (532- and 1064-nm wavelengths) which is capable of determining cloud phase based on the differing backscatter of ice crystals and liquid droplets (Wang et al., 2012). CALIOP's short wavelengths attenuate quickly and only penetrate clouds with relatively low optical depths, ∼3 or less (Chepfer et al., 2010), so CALIOP on its own is not capable of providing information on moderate to heavy snowfall or snowfall beneath liquid cloud layers. The CPR's longer wavelength, however, can generally penetrate all Arctic clouds to detect underlying precipitation (Battaglia and Delanoë, 2013). It is the combined skill of these instruments that allows for this study.

CloudSat experienced a battery failure in 2011, causing the CPR to only provide data for daytime overpasses. Due to the high latitude position of the GrIS, this malfunction has a seasonal impact, rather than a daily one. In summer, the GrIS experiences nearly constant solar illumination so there is no difference in the pre- and post-2011 summertime data collection. While there is a reduction in the post-2011 wintertime Arctic data, it is not eliminated completely. The CPR continues to function for some minutes as it passes into the darkness of boreal winter, resulting in the collection of approximately half of the wintertime Arctic observations collected prior to the battery malfunction (Skofronick-Jackson et al., 2019).

The satellite data used in this study is referred to in terms of overpasses and footprints. An overpass is an individual flyover of the GrIS, and comes from a single granule of CloudSat data — one orbit around the Earth (roughly 1.5 hours). Each CloudSat footprint has a horizontal resolution of approximately 1.4 x 1.7 km at the surface and a vertical bin resolution of approximately 240 m. Because of its shorter wavelength, ∼12 CALIOP footprints fit within a single CPR footprint. The CALIOP cloud phase information has therefore been scaled to the CPR resolution in the below described data products. In this study we use all available footprints where both the CPR and CALIOP were functioning, which includes data between January 2007 and August 2016.

This study is primarily based on three data products produced by the CloudSat Data Processing Center: 2C-SNOW-PROFILE, 2B-CLDCLASS-LIDAR, and 2B-GEOPROF (hereafter 2CSP, 2BCCL, and 2BG, respectively). These products have all been extensively described elsewhere so the reader is directed to the citations provided below for algorithm and validation details. All granules available in the R05 CloudSat data product release as of May 2019 are used - no years or months were excluded. In

total this research includes more than 17 thousand overpasses of the GrIS, consisting of 14.7 million total individual footprints, 2.4 million of which contained snowfall (Fig. 1).

The first step in our analysis is to obtain snowfall frequency and rate from 2CSP, a radar-only product that uses CPR reflectivity information from the lowest clutter free bin to estimate surface snowfall rates (Wood and L'Ecuyer, 2018). The CPR cannot directly observe snowfall at the surface because of ground clutter — the bright surface return overwhelms the detector and creates a blind zone in the ~1-1.2 km closest to the surface. To make a surface rate estimation, 2CSP relies on the connection between precipitation-sized particles aloft and snowfall at the surface; the downward snow mass flux retrieved at the top of the blind zone is assumed to reach the surface (discussed in Section 3). Studies comparing 2CSP to surface data have validated this connection, specifically in the polar regions (Milani et al., 2015; Norin et al., 2015; Palerme et al., 2016). The minimum detectable rate for 2CSP is ~0.0005 mm/h.

We define a footprint as snowing if it has a non-zero 2CSP snowfall rate in the lowest clutter free bin, unless that bin is flagged as possibly contaminated by ground clutter. Potential contamination is indicated by the third bit in the 2CSP status flag, which is set when there is a large difference between the snowfall rate in the lowest clutter free bin and the bin immediately above. This occurs naturally if there is a very shallow snowfall event such as lake effect snow, where the precipitation-sized particles are confined to ~1-1.5 km of the surface, or it means that lowest bin expected to be clutter free is actually contaminated by surface return. Contamination is most prevalent in regions of steep, icy topography where the digital elevation map used to determine the surface level does not exactly match conditions at the time of the overpass (Bennartz et al., 2019). Palerme et al. (2019) showed that the edges of the GrIS are particularly prone to clutter in the R04 version of 2CSP, but the updated elevation map in R05 has reduced the number of contaminated pixels. In this study, when 2CSP identifies a satellite footprint as having potential contamination in the lowest bin that should be clutter free, we take the snowfall rate from the level immediately above, consistent with the methods of Palerme et al. (2019) and Milani et al. (2018). Bennartz et al. (2019) also highlighted the issue of surface contamination in GrIS snowfall estimates using 2CSP, but approached a solution by creating a completely new rate based on reflectivities aloft.

In a given footprint, if 2CSP indicates snowfall at the surface, we then obtain cloud phase for that footprint from the 2BCCL. The sensitivities of both the CPR and CALIOP are leveraged by 2BCCL to determine phase (Wang et al., 2012): the lidar is particularly sensitive to cloud liquid layers while the CPR provides additional ice crystal information that the lidar may miss due to attenuation. The relative strengths of the radar and lidar backscatters are distinct for each cloud phase: ice clouds produce weak to moderate lidar and strong radar backscatter; liquid clouds show strong lidar and weak radar backscatter; and the backscatter for both instruments is strong for mixed-phase clouds. 2BCCL gives each vertically contiguous cloud a single phase (ice, liquid, or mixed), regardless of how the particles within the cloud are distributed. If there are multiple cloud layers in a given column, we take the phase of the lowest cloud layer. Our liquid-containing classification (CLW) includes both 'liquid' and 'mixed' flags while our ice classification (IC) uses only 'ice'.

Finally, we use a second radar-only product, 2BG (Marchand et al., 2008), to further characterize the clouds producing snowfall by looking at reflectivity properties. From 2BG we obtain the height and magnitude of radar reflectivity factor, Ze, in the vertical column and also the vertically integrated reflectivity, $Z_{path}$, discussed below.

For the regional maps, all A-Train data were binned to a ∼0.94° latitude by 1.25° longitude grid (consistent with McIlhattan et al. 2017). In addition to gridding the observations, we have also collected satellite footprints made within each GrIS drainage basin as defined by Zwally et al. (2012) and shown in Fig. 1. The basins each have consistent surface slope relative to atmospheric advection, enabling us to look at snowfall characteristics in large regions that are more meaningfully homogeneous than grid boxes (Zwally et al., 2012).

## 2.2 $Z_{path}$

In this work we use column-integrated reflectivity ($Z_{path}$, mm$^6$ m$^{-2}$) as a proxy for the ice mass characteristics of the cloud. $Z_{path}$ is a relatively simple measurement related to the amount of hydrometeor backscatter (defined as $Z_{int}$ in Kulie et al., 2010; Pettersen et al., 2016). It is defined as:

$$Z_{path} = \int_{H_{CB}}^{H_{CT}} Z_{CPR}(z)dz, \tag{1}$$

with $H_{CT}$ and $H_{CB}$ as the cloud-top height and base, respectively, and $Z_{CPR}$ is the CloudSat CPR radar reflectivity factor at a given height, $z$. Cloud boundaries come from 2BCCL and the reflectivities between those boundaries come from 2BG. The 2BG reflectivity factors are converted from the provided $dBZe$ to $Ze$ before integrating, then from $Z_{path}$ to $dB(Z_{path})$ for plotting and discussion (consistent with Kulie et al. (2010)).

## 2.3 Ground-based Data

The ground-based observations of snowfall, cloud phase, and radar reflectivity used in this study were collected as part of the ongoing Integrated Characterization of Energy, Clouds, Atmospheric State and Precipitation at Summit (ICECAPS) project (Shupe et al., 2013). Summit Station is located at 72°36' N, 38°25' W and is denoted with a white star on all GrIS maps in this work (e.g. Fig. 1).

Surface detected snowfall events at Summit were defined and segregated into IC and CLW events using the novel method developed and detailed in P18. P18 leveraged differences in absorption and scattering properties of cloud liquid and ice in MWR measurements to separate the two precipitation regimes. The IC and CLW surface based snowfall data are for the period 2010-2015. The data product as well as technical details are available in the National Science Foundation Arctic Data Center archive (Pettersen and Merrelli, 2018).

In Section 3, we use averaged reflectivity measurements from ICECAPS's millimeter wavelength cloud radar (MMCR) to estimate the detectability of surface snowfall events from space by the CPR. The MMCR is a vertically pointing, 35 GHz, Doppler pulsed radar (Moran et al., 1998) that is sensitive to both ice and liquid hydrometeors.

To convert the MMCR reflectivity native resolution to CloudSat-like footprints we use time averaging and thresholds that closely mimic the CPR and its algorithm: a height range of 960-1200 m above ground level (AGL), equivalent to the standard height of bin 5 of the CPR used in the 2CSP algorithm over land; and a time average of 300 s, which at a moderate wind speed

of 5 m s$^{-1}$ is equivalent to the horizontal CPR footprint of ~1.5 km. Most wind speeds detected during Summit snowfall events are faster than 5 m s$^{-1}$ (P18), however, using faster wind speed thresholds (shorter time averaging) results in more detected cases so this slower threshold was chosen to provide a conservative estimate of detected cases. Missing MMCR reflectivities are excluded from the average, while clear bins are included.

5    The 1-minute resolution snowfall regime data (Pettersen and Merrelli, 2018) is sampled to match the MMCR time averaging period. If snow occurred (defined in P18 as Precipitation Occurrence Sensor System (POSS) power unit >2) for any time during the sample, the sample counts as a snow occurrence, even if for the majority of the averaged time no snow was falling. For sample mean snowfall rates, the POSS rate was averaged over the sample with missing values omitted and values associated with POSS power unit <2 included as zeros.

## 10  2.4  Reanalyses

Similar to P18, in Section 4.3 we use reanalyses to examine the atmospheric circulations associated with IC and CLW events for various GrIS regions. The European Centre for Medium-Range Weather Forecasts (ECMWF) provides the global reanalysis product ERA5 (C3S, 2017), from 1950 to present. ERA5 contains hourly data with a latitude and longitude spatial resolution of 0.25° x 0.25°.

In Section 4.3, we plot maps of the mean and anomaly of the 500 mb geopotential height (GPH) and winds associated with snowfall events for the two regimes. The means are composites of the entire region using hourly ERA5 data nearest in time to the selected snowfall events. Climatological anomalies are generated by subtracting the long-term ERA5 monthly mean (1979-2018) from the hourly ERA5 data nearest in time to the selected snowfall events.

## 3  Quantifying Snowfall Detectability from Space

In this section we investigate the reliability of CloudSat's CPR in detecting GrIS snowfall. Radar-derived snowfall rate estimates are dependent on assumptions about properties including ice particle shape, properties that are variable in space and time (Kulie et al., 2010). While the 2CSP snowfall rate is impacted by these assumptions, snowfall detection and resulting frequencies are independent of them. If the CPR detects precipitation size particles immediately above the blind zone, it is a good indicator that snow is falling at the surface (Boening et al., 2012; Milani et al., 2018; Palerme et al., 2019; Bennartz et al., 2019).

The ICECAPS instrument suite at Summit provides a unique opportunity to look at IC and CLW cases from below, which can provide insight into what is seen from above. When comparing 200 m and 700 m AGL snowfall reflectivities from an ICECAPS radar, Castellani et al. (2015) found evidence of growth — the reflectivities at 200 m AGL were larger than 700 m AGL on average — suggesting that towards the surface there is an increase in particle masses, an increase in number concentration, or a shift from small particles to large particles in the size distribution (or a combination thereof). Since an 30  increase in reflectivity can arise from one or more of these different processes, when we say "growth" throughout this work, we are not specifically implying particle mass increase, but the collection of snow property changes that can influence reflectivity. Castellani et al. (2015) showed the distribution of reflectivity differences between 200 and 700 m AGL have both positive and

negative values, meaning that while on average snowflakes at the top of the blind zone likely underwent growth as they fell, it was not guaranteed. McIlhattan et al. (2017) examined the presence of clouds containing super-cooled liquid over Summit and the frequency with which they precipitated, finding that the 2CSP and 2BCCL (R04 versions for 2007-10 only) matched well with the surface observations from Miller et al. (2015). Figure 6 of P18 further supports the idea that snowfall cases should be

detectable from space, since reflectivities greater than 0 dBZ occur frequently above the blind zone: up to 3 km AGL for IC snowfall cases and 2 km AGL for CLW cases. However, the MMCR reflectivities shown in P18 cannot be directly compared to the CPR due to differing space and time averaging.

Previous papers have mentioned that due to the blind zone of the CPR, a number of snowfall events are likely missed (e.g. Maahn et al., 2014; Palerme et al., 2019; Bennartz et al., 2019). We aim here to quantify that number for our two

snowfall regimes. By averaging the Summit MMCR data for IC and CLW cases, we create a CPR-like vertical profile and use coincident POSS measurements to define cases as snowing or not. Averaging and instrument details can be found in Section 2.3. For profiles with snow occurrence, the sample is considered missed by the CloudSat-like MMCR observations if the radar reflectivity in the selected CloudSat vertical bin is below the -15 dBZe threshold as defined in the 2CSP algorithm. Despite the differing wavelengths of the CPR and MMCR (frequencies of 94 and 35 GHz, respectively), at the snowfall defining threshold

of -15 dBZe their reflectivities are comparable. For such small reflectivities, in most cases the ice particles are small such that the reflectivity is in the Rayleigh regime for both wavelengths.

The results of this comparison are summarized in Table 1. Note that when the MMCR was averaged over time, sometimes more than one type of Summit snowfall event (IC, CLW, or indeterminate) were blended together. For clarity, we are only showing results for the combined total, IC-only, and CLW-only scenes. In IC-only averaged scenes, all included snowfall

events contained only fully-glaciated ice clouds. In CLW-only averaged scenes, cloud liquid was present in each snow event, though to be clear the CLW clouds are almost always mixtures of both supercooled-liquid water and ice.

Out of 20,516 total snowfall events identified in the averaged P18 dataset, 22 % of the events would have been undetected by the CPR. When looking at the 9,777 CLW-only snowfall events, the missed fraction goes up to 25 % and for the 3,545 IC-only events the missed fraction goes down to 5 % (the remaining 7,194 cases are mixed or indeterminate). The mean snowfall rate

reported by the P18 dataset for the missed events is consistently about half the rate of the detected cases, meaning that the CPR is missing the lighter of the events overall, and within both of the regimes. Broadly, these results indicate that the CPR is sensitive enough to detect nearly all of the IC-only cases (95 %) as they appear at Summit, but has more difficulty with the Summit CLW-only cases, capturing a smaller majority (75 %).

Bennartz et al. (2019) showed that 2CSP underestimates snowfall accumulation near Summit Station relative to stake field

and MMCR estimates. P18 showed that CLW cases are responsible for slightly more than half of accumulation at Summit. Our results here indicate the CPR is likely missing ∼25 % of CLW cases, which could mean that the low 2CSP accumulation bias at Summit is an issue of missing snowfall cases entirely, rather than an underestimate of rate as was suggested in Bennartz et al. (2019). However, it is not clear that the missed cases identified here should result in large scale biases in snowfall frequency or total accumulation values. Maahn et al. (2014) compared snowfall values derived from CloudSat and derived from ground

based radar at sites in Norway and Antarctica, finding that the competing effects of shallow snowfall not being seen by the CPR

and virga that was flagged as snowfall by the CPR though did not reach the surface resulted in CPR derived frequency being different by $\pm 5$ % and the total amount being underestimated by 9–11 %, relative to ground based values. Similarly, Ryan et al. (2020) found that CPR derived snowfall rates correlate well with precipitation gauges at two locations on the surface of Greenland, and with CPR rates for two particular snowfall events coming within $\pm 9$ % of the precipitation gauge value. The
following results are not modified based on CPR under-detection, but the implications are touched on in the conclusion.

## 4  IC and CLW Regime Characteristics

### 4.1  Snowfall Frequency and Accumulation

The frequency of detected snowfall events varies both regionally and seasonally over the Greenland Ice Sheet (GrIS). There is a north-south gradient in the annual frequency map of all snow events (Fig. 2a), with frequency increasing towards the southern
end of the GrIS. The highest concentration of snowfall observations is along the southeastern coastline. This is consistent with previously documented heavy snowfall in the area, with studies attributing it to the region's steep orography and interaction with paths of North Atlantic storms (e.g. Schuenemann et al., 2009; Hakuba et al., 2012; Berdahl et al., 2018). When we partition the annual snowfall into cases coincident with ice clouds (IC) and clouds containing liquid water (CLW) (Fig. 2, b and c, respectively) the GrIS snowfall frequency is clearly dominated by IC events. There is an east-west gradient in the regime
fraction, with more CLW cases along the western side of the GrIS than the eastern.

In wintertime (defined here as October through May, consistent with P18), there is very little snowfall in the northern GrIS and an even stronger north-south gradient compared to the annual distribution (Fig. 2d). The concentration of events along the southeastern coastline is also more prominent, with snowfall occurring up to 40 % of the time. This is consistent with the wintertime high concentration of cyclone centers and increased cyclone intensity off the southeastern GrIS coastline, found
by Zhang et al. (2004) using reanalyses. IC events (Fig. 2e) make up nearly 100 % of the wintertime snowfall observations over most of the GrIS, with the exception of western Greenland where CLW approaches 50 % of the cases in some grid boxes nearest the central coast (Fig. 2f).

In the summer months (defined here as May through September, consistent with P18), the north-south gradient is gone, with a fairly consistent 20-30 % snowfall frequency over the GrIS (Fig. 2g). The precipitation occurring at the edges and outside of
the ice sheet is predominantly rainfall during this season, and since rain is excluded from this study the frequency over the coast and ocean is reduced. CLW cases make up ∼50 % of the snowfall frequency over much of the ice sheet in summer months (Fig. 2i), which is consistent with what P18 observed at Summit Station. The southeastern coastline, however, remains more influenced by IC snowfall even in summer. The east-west gradient in regime fraction is distinct in summer, with more CLW along the western side of the ice sheet (reasons for this east-west regime divide are examined in Section 4.3). In the summer
30  months, the GrIS receives ∼83 % of its annual incoming solar insolation (calculated using 2B-FLXHR-lidar data (Henderson et al., 2013)). So while the IC events clearly occur more often annually, the CLW events are equally important for brightening the GrIS and increasing the surface albedo during the months of intense downwelling shortwave radiation. While not the focus of this work, it is important to remember that CLW events have the competing surface effects of enhancing albedo with snow

while the liquid bearing clouds also trap additional terrestrial radiation, potentially enhancing melt (Van Tricht et al., 2016; McIlhattan et al., 2017).

The surface of the GrIS ranges in elevation from near sea level to above 3,200 m. The topographical variation results in large spatial gradients in near surface atmospheric conditions (temperature, available moisture, etc.) which, in turn, influence snowfall characteristics. In Fig. 3, we separate snowfall frequency by elevation and season for two regions: the southeastern GrIS, where snowfall has the highest frequency, and the western GrIS, where CLW snowfall makes up the largest fraction of snowfall events. In winter in the southeastern GrIS (Fig. 3a), total snowfall (black bins) occurs ~35 % of the time below 1,500 m, and decreases dramatically above that elevation. This behavior results from two trends: (1) CLW events steadily decrease (blue bins) with increasing elevation and (2) IC events increase with elevation up to 1500 m but then decline as elevation increases further (red bins). The CLW decrease is consistent with the strong temperature gradient that occurs with elevation, where at high elevations the wintertime temperatures and humidity do not favor liquid containing clouds (Shupe et al., 2013). Snowfall in the western GrIS in winter (Fig. 3c) is less frequent in all elevations except the highest bin (3,000 – 3,500 m) compared to the southeastern GrIS. Western winter CLW events decrease while IC increase with increasing elevation (Fig. 3c), similar in behavior to the southeastern GrIS, however the relative magnitude of changes are such that the total snowfall in the western GrIS increases with increasing elevation up to 3,000 m. In summer, the snow frequency characteristics are consistent between the southeastern and western regions (Fig. 3, b and d): total snowfall frequency generally increases with elevation, IC events increase with elevation, CLW events are more frequent than IC events at the lowest elevation, and CLW events maintain a relatively consistent frequency at all elevations. The steady frequency of CLW events in summer, independent of elevation and distance from the GrIS coastline, may be partially explained by the fact that super–cooled liquid containing clouds in the Arctic are resilient and long lasting, able to replenish moisture through cloud top processes even while producing snowfall at their base (Morrison et al., 2012).

Snowfall rates decrease with increasing elevation for IC and CLW events in both the southeastern and western GrIS and in both winter and summer (Fig. 4). Snowfall rates in the southeastern GrIS are higher than the western GrIS for both IC and CLW. Except for elevations below 1000 m in the southeastern GrIS, snowfall rates are higher in summer than in winter, consistent with increased temperatures and moisture availability. In those low elevations of the southeastern GrIS, rainfall is known to occur in the summer months (Lenaerts et al., 2020). Since rainfall is screened out in our analysis, the total precipitation rate (rainfall + snowfall) in summer is potentially higher than in winter for the two regimes.

Snowfall accumulation is the largest positive term in the surface mass balance of the GrIS (e.g Jakobson and Vihma, 2010; Mottram et al., 2019). The estimate of mass added to the GrIS by snowfall by season and regime is shown in Fig. 5. The mean annual accumulation for the study period is 399 Gt $yr^{-1}$ (Fig. 5a) which is distributed nearly equally between winter (198 Gt $yr^{-1}$, Fig. 5d) and summer (201 Gt $yr^{-1}$, Fig. 5g). However, by our definition summer represents only five months compared to winter's seven, meaning that the intensity of summer snowfall is greater on average. The total annual snowfall in Fig. 5a is consistent with the results from Bennartz et al. (2019), with the highest fraction of GrIS snowfall occurring in basins 8.1 and 6.2. respectively. The largest accumulation in winter occurs in the basins along the southeastern coastline (Fig. 5d), in agreement with other studies highlighting more and/or stronger cyclones (Zhang et al., 2004) and precipitation (e.g. Vihma

et al., 2016; Berdahl et al., 2018) in that region in winter. In summer the largest accumulation is in the basins of the central west (Fig. 5g), a result of the combination of both increased snowfall rate in the region and the relatively large area of these basins. Snowfall from IC events makes up ∼80 % of the total annual accumulation by mass, ∼88 % of the winter, and ∼71 % of the summer accumulation (Fig. 5, b, e, and h, respectively). While there is some seasonal variation in the accumulation and distribution between regimes, it is clear that in all basins the majority of the snowfall mass comes from IC events. The accumulation by individual basin is summarized in Table 2.

Previous estimates for GrIS mean annual accumulation have generally been higher than this study, with different models, configurations, and reanalyses ranging from ∼581 - 899 Gt yr$^{-1}$ (Cullather et al., 2014) and the recent CloudSat observational study, Bennartz et al. (2019), reported 586 ± 129 Gt yr$^{-1}$. As discussed previously, models and reanalyses rely on observations for constraints, and over the GrIS those have historically been sparse. The CPR derived snowfall rate in Bennartz et al. (2019) had a correction (relative to 2CSP) to increase high elevation rates to more closely match Summit observations with the assumption that there would be little effect outside of high elevations because snowfall was expected to be associated with higher reflectivities. While this is likely true for IC cases, the following analysis demonstrates that CLW clouds are consistently thinner geometrically and with low IWP over the full ice sheet. Our GrIS accumulation estimate is likely biased low because the CPR is missing ∼25 % of CLW snowfall cases (as discussed in Section 3), but it is not clear that tuning all high elevation snowfall rates to one particular location will improve our larger scale evaluation and thus we present 2CSP rates as they are.

A histogram of the rates for all observed snowfall (Fig. 6a, top) illustrates that the vast majority (note the log scale on the y-axis) of snowfall observations are very light. Just over 92 % of the snowfall observed is contained in the first bin, which includes snowfall rates of ≤ 0.41 mm hr$^{-1}$ liquid water equivalent. Snowfall is frequent in both seasons, with winter and summer time periods each accounting for roughly half of the snowfall-containing satellite footprints. The winter histogram (Fig. 6 b, top) looks much the same as the annual, though has slightly steeper drop-off from first to the second bin, indicating that winter snowfall is often lighter, fitting with the general scarcity of available atmospheric moisture during these months. The summer histogram (Fig. 6c, top), on the other hand, shows a smaller decrease between the first and second bins, consistent with generally more summertime atmospheric moisture allowing for increased ice particle formation and/or growth. The slope of the distributions between the two seasons is distinct, with summer having an overall steeper decline and fewer observations over 6 mm hr$^{-1}$ compared to winter. This means that while the common summer events are snowing at slightly higher rates on average, it is in winter that the less frequent, highest-intensity snowfall occurs. Jakobson and Vihma (2010) found a similar relationship using reanalysis data in the Arctic, showing winter having lower precipitation rates than summer overall, but the annual precipitation maximum occurring in winter along the southeastern GrIS coastline. They attributed the regionally strong winter snowfall to the strength and position of the North Atlantic cyclone tracks.

The distribution of snowfall events between the two regimes is stark. In the annual (Fig. 6a, bottom), IC observations (red) are more frequent at all snow rates. The largest fraction of CLW events (blue, ∼32 %) occurs at the lightest snowfall rates and the fraction decreases rapidly, with all events greater than 6 mm hr$^{-1}$ produced by ice clouds, consistent with the findings of P18 at Summit. In winter (Fig. 6b, bottom) the CLW fraction decreases to ∼18 % for the lightest events and IC are responsible for greater than 95 % of the observations of snowfall >2 mm hr$^{-1}$. CLW and IC produce nearly the same number of light events

in summer (Fig. 6c, bottom), and CLW has a larger share of the moderate events than in winter. However, in both summer and winter, the heaviest snowfall is produced by IC events.

The motivation for this study was to determine if the analysis in P18 at Summit Station could be expanded to the full GrIS, and if so, find out how the regime characteristics compare. To compare with the point source ground measurements from P18, we selected only satellite observations made within 100 km of Summit Station (the starred circle in Fig. 1). Figure 7 illustrates the annual cycle of regime cloud frequency: the fraction of IC events and the fraction of CLW events out of all the snowfall events observed. The CLW fraction for both the satellite (solid blue line) and the ground-based (dashed blue line) observations have close agreement, particularly in the summer months. The satellite CLW fraction is lower year round than the surface observations, which fits with the CPR missing ∼25 % of CLW events, as discussed in Section 3. The closer match between ground- and space-based observed fractions in summer could be due to the higher cloud water content improving detectability from space. The IC events (red lines) from the two platforms follow a similar pattern in the summer months, though the ground-based fraction is smaller. During the winter there is a clear majority of near-Summit IC events observed from space. The surface observations, on the other hand, show a drop off in IC snow during winter, and at the same time have a minimum in CLW events. This results likely from the third, "indeterminate", category present in ground-based Summit observations in P18, but not in 2BCCL, being more prevalent in winter.

A key takeaway of Fig. 7, beyond general agreement between the two instrument platforms on the relative frequency of regimes, is the important role CLW events play in brightening and adding mass to the surface of the central ice sheet during the summer months. CLW events are roughly double the number of IC in July, and continue to dominate frequency in August, the highest month of accumulation at Summit (Bennartz et al., 2019).

## 4.2 Cloud Characteristics

The annual cycle of geometric cloud depth for IC and CLW snowfall events (Fig. 8a) demonstrates that these regimes are consistently physically distinct in all months of the year. The CLW clouds are on average much shallower than the IC clouds, which is consistent with previous understanding of these regimes (e.g. Morrison et al., 2012; P18). The mean IC geometric depth hovers around 4 km with two broad maxima: one centered around January and one around August. The mean CLW geometric depth is between 1.5 and 2.5 km, with a single peak in July-August. Looking at the monthly average and standard deviation (solid line and shaded region, respectively) for the full GrIS, there is no overlap between the two regimes in any time of year.

At all elevations, the mean geometric cloud depths are larger for IC events than CLW events (Fig. 9, a and b). The maximum for both regimes occurs at the lowest elevations and clouds tend to become shallower with increasing elevation. There is some variation regionally, with the southeastern GrIS (filled circles) showing a stronger dependence of cloud depth on elevation and a generally larger cloud depth than the western (filled triangles) or northern (filled diamonds) GrIS. Differences in these regions for the two regimes are explored further in Section 4.3.

Since the geometric cloud thickness is so distinct between the two regimes, it follows that the cloud water content will also be different. In this work we use $dB(Z_{path})$ as a proxy for IWP (see method section 2.2). Looking at the annual cycle of

$dB(Z_{path})$ for the full GrIS (Fig. 8b), the IC monthly averages show one main peak between May and October. IC $dB(Z_{path})$ has particularly small inter-annual variability June through September, the months with highest $dB(Z_{path})$. In contrast, the CLW $dB(Z_{path})$ has a broader and shallower summer peak and larger year to year variation (shown by the broader shaded region and spread of monthly markers) compared to IC events.

5  For the clouds observed within 100 km of Summit Station, the annual cycle shows that the IC events are consistently thicker geometrically than CLW events (Fig. 10a), though with more inter-annual variability compared to the full GrIS (illustrated by the relatively larger shaded region) and no discernable annual cycle. Using ground based remote sensing, Miller et al. (2015) also found no clear annual cycle in integrated thickness for clouds above Summit Station. The near-Summit satellite observations of $dB(Z_{path})$ (Fig. 10b) have increased variability between years relative to the full GrIS. Though the IC clouds

10 still have higher $dB(Z_{path})$ than CLW year round, both regimes have much smaller monthly mean $dB(Z_{path})$values near-Summit than for the full ice sheet. This implies decreased IWPs for both regimes near Summit relative to the rest of the GrIS, consistent with the greater distance from moisture sources.

  The transition from large to small $dB(Z_{path})$ occurs gradually from the low lying coastal areas to the peak of the ice sheet. Figure 9, c and d, show that for the whole GrIS and individual regions, $dB(Z_{path})$ decreases consistently with elevation for

15 IC and CLW events. In summer, $dB(Z_{path})$ is higher at all elevations than in winter. Regionally, the southeastern GrIS has the highest $dB(Z_{path})$ at all elevations, followed by the western region, and the northern GrIS has the lowest.

  It is useful to look also at the distribution of individual snowfall events to understand the character of the clouds that make up the mean. The top row of Fig. 11 is a collection of histograms containing all of the observed snowfall over the GrIS for the entire study period. From left to right we have: annual, winter, and summer time periods (a, b, and c, respectively). Overall,

20 most observations of snowfall (70 %) over the GrIS are coming from IC events; however, in the summer months, close to half (45 %) of the snow observed is produced in CLW events. Similar to the plot of the annual cycle, Fig. 11a, shows that overall IC and CLW snowfall event clouds have distinctly different geometric depths. However, the overlap in their distributions indicate that there are some individual IC events that are shallower and CLW that are deeper than is implied in the annual cycle plot. Each regime histogram is individually normalized to better focus on and compare the shapes of the distributions, rather than the

25 magnitudes. The CLW clouds are remarkably invariant across seasons, with a narrow distribution and a peak between 1 and 2 km in geometric thickness (Fig. 11, b and c). The tail of the CLW distribution changes between winter and summer — summer has a longer tail to the right of the peak, responsible for the slight increase in mean thickness for those months seen in Fig. 8a. The IC clouds have a broader range of geometric depths than the CLW, with a wide peak in the annual distribution between 2 and 5 km geometric thickness that becomes slightly thinner in winter (2-4 km) and thicker in summer (3-6 km). There is also

30 a change in the skewness of the distribution, with a positive skew (peak to the left, tail to the right) in the winter and a little to no skew (peak centered in the range of measurements) in summer. The difference in the shapes of the distributions means that the average cloud depth does not shift as much between seasons as the peaks in the distribution would imply. The distribution of cloud depths near Summit (Fig. 11, bottom row) are noisier, but demonstrate consistency with the full GrIS results in both shape and seasonal characteristics.

The histograms of $dB(Z_{path})$ (Fig. 12) highlight distinct seasonal behavior for the two regimes. The CLW events are again quite invariant between the annual, winter, and summer plots for both the full GrIS (top) and within 100km of Summit (bottom). GrIS-wide, the CLW has a sharp peak at ~0 $dB(Z_{path})$, positive skew, and overall similarly shaped distributions during all three time periods. The GrIS IC event distributions have broader peaks that are consistently at larger $dB(Z_{path})$ values than the CLW peak, and much higher values (~10 $dB(Z_{path})$) in summer. This indicates higher IWP year-round in the GrIS IC snow events compared to CLW events, and an increased summer ice path for the IC events, coinciding with the peak in the annual cycle plot (Fig. 8b). The skewness of the IC events is again marked, with the annual having no strong skewness and winter and summer showing opposite skewness, positive and negative, respectively. This means that the $dB(Z_{path})$ of the most commonly present cloud (the mode) in the two seasons is more disparate than shown by the mean in the annual cycle. The near Summit distributions are again noisier compared to the full GrIS, but are quite similar in overall behavior, though with relatively fewer values above ~10 $dB(Z_{path})$.

Taking the cloud geometric thickness and $dB(Z_{path})$ together, another distinction becomes clear: the GrIS CLW events exhibit relatively constant characteristics throughout the year, while the GrIS IC events are more seasonally dependent, within limits. IC events have a distinct annual cycle for both cloud geometric depth and $dB(Z_{path})$, but the variability (shown by the shaded standard deviation) within that cycle is generally no larger than the CLW variability, and in the summer in particular the variation is smaller. The summertime $dB(Z_{path})$ increase in IC events is accompanied by only a small increase in geometric thickness, meaning the clouds are denser during this period.

While $dB(Z_{path})$ gives an estimate of the total ice content of a cloud, the distribution of that ice in the vertical profile can give insight into hydrometeor growth tendencies. As a reminder, by "growth" we refer to an increase in particle masses, an increase in number concentration, or a shift from small particles to large particles in the size distribution (or a combination thereof). Figure 13 contains two-dimensional (2D) histograms of CPR reflectivities as a function of height for each regime. The composite of all observed IC snow events (Fig. 13a) shows increasing reflectivity toward the surface, indicating growth from the top of the deep clouds moving down the column to the top of the blind zone. The IC histograms in the winter and summer (Fig. 13, c and e, respectively) have narrower distributions for given heights compared to the annual, with generally higher reflectivities in summer but showing consistent growth patterns in both seasons.

The CLW snowfall does not have as defined a relationship between height and reflectivity (Fig. 13b), shown by the rounded distribution. The CLW winter and summer (Fig. 13, d and f, respectively) histograms have similar, round distributions and reflectivity spreads, though the peak in height is slightly higher in winter (~2.5-3 km AGL) than in summer (~2 km AGL). Unlike the IC distribution, the CLW shape does not display a discernible growth pattern. Both the IC and CLW results are consistent with what P18 found using the MMCR from the ground.

### 4.3 Associated Atmospheric Circulations

While many local factors influence when and where snowfall occurs over the GrIS (topography, surface type, temperature, etc.), variations in atmospheric circulation have been determined to be the primary control on GrIS snowfall accumulation (e.g. Alley et al., 1993; Kapsner et al., 1995; Chen et al., 1997). Knowing the large-scale meteorological conditions that are

coincident with each snowfall regime can help better constrain both the present day mass balance of the GrIS as well as predict how it might change in the future. In this section, we examine the atmospheric circulations associated with regime snowfall for four GrIS regions: near-Summit, southeastern, western, and northern.

First we look at near-Summit cases to find the coincident atmospheric conditions that are able to bring IC and CLW events all the way to the center of the GrIS. To be included as a case for one of the regimes, an individual Summit overpass needed to contain a minimum of 10 contiguous snowfall footprints (equivalent to ∼15 km along-track) within 100 km of Summit Station, and of those footprints a minimum of 90 % had to be of that regime type. We use only the most intense 50 % of IC and 50 % of CLW events identified by CloudSat/CALIPSO. To rank the strength of observed events, we take a cumulative sum of surface snowfall rate for a single overpass within a basin. Thus, both large-scale, light snowfall and small-scale, heavy snowfall are included in the top 50 %. This selection process results in 159 IC cases and 43 CLW near-Summit cases.

The mean 500mb geopotential heights (GPH) for near-Summit IC events (Fig. 14a) display a trough-ridge feature with a gradient bisecting the GrIS. The mean winds close to Summit are coming from the south-southeast. The 500 mb GPH anomaly (Fig. 14c) shows a dipole with higher than average heights in the North Atlantic and lower than average over the western GrIS and Baffin Bay. The anomalous winds are strong relative to the mean and come from the south-southeast. These conditions are similar in character and magnitude to what P18 found when looking at IC events at Summit Station. The IC 500 mb mean and anomalous GPH patterns are consistent with low-level convergence advecting warm, moist air from the North Atlantic ocean surface vertically though the column and north over the steep southeast coast of Greenland. These conditions bear strong resemblance to synoptic conditions often credited with GrIS snowfall (Chen et al., 1997; Serreze and Barrett, 2008; Rogers et al., 2004; Schuenemann et al., 2009).

The regional map of mean 500 mb GPH for near-Summit CLW cases (Fig. 14b) indicates calm conditions, showing relatively uniform heights over the GrIS and with low wind speeds around Summit coming from the south-southwest. The main feature of the CLW GPH anomaly (Fig. 14d) is much higher than average heights over the entire GrIS. Hanna et al. (2016) identified that persistent high pressure anomalies are consistent with increased GrIS precipitation in reanalyses in the western and central regions. These CLW conditions are again consistent with what was found in P18. This overall picture of quiescent flow and large-scale subsidence is known for maintaining Arctic mixed-phase clouds (Morrison et al., 2012). While in this work we are focused on precipitation production, it is worth noting that these conditions also have the potential to enhance melt over the GrIS through radiative forcing (Van Tricht et al., 2016).

The strength of our satellite approach is that we can look beyond Summit station to extend the P18 surface-based analysis and examine conditions coincident with regime snowfall in areas without ground-based observatories. We start by looking at cases of snowfall in the northern GrIS, defined here as basins 1.1, 1.2, 1.3, and 1.4. Shupe et al. (2013) and Castellani et al. (2015) found essentially no snowfall at Summit associated with northerly surface wind components. Similarly, P18 saw negligible northerly surface winds with IC cases and a very small component for CLW cases, though northerly surface winds did occur outside of precipitation events. This hints that the air-masses responsible for northern GrIS snowfall do not move on towards the central GrIS.

Choosing the strongest 50 % of overpasses for each snowfall regime during the study period, we have 1,125 IC cases and 452 CLW cases making up the composite maps for the northern GrIS (Fig. 15). There are more cases in the northern GrIS than any other region we examine because of the concentration of satellite overpasses (see Fig. 1), not because it snows more frequently there. The IC mean 500 mb GPH map (Fig. 15a) contains a trough to the west of the GrIS and very calm upper level winds in the northern GrIS. The GPH anomaly map for IC cases (Fig. 15c) has a dipole centered over the GrIS with higher than average heights to the west of the GrIS and a low centered on the northeastern coast, with the anomalous winds coming from the north into our basins of interest. In an analysis by Chen et al. (1997) looking at synoptic causes for GrIS precipitation, it was found that high precipitation in the northern GrIS in 1987-88 was associated with a mean cyclone located in the Arctic Ocean close to the northeast coast of Greenland. The low anomaly in IC 500 mb GPH seen in Fig. 15c is suggestive of a mean cyclone in that location.

The northern GrIS CLW cases are associated with a markedly different circulation pattern. Much like the near-Summit CLW cases, the mean 500mb GPH for northern GrIS CLW cases has relatively uniform heights and low wind speeds. There is a strong anomalous ridge centered over Baffin Bay extending over the full GrIS in the GPH anomaly plot (Fig. 15d), with high anomalous northerly winds similar to the IC cases. The high anomalous winds moving over the northern GrIS are pointed southeast towards the center of the ice sheet, however the actual mean wind speeds present there are very low and coming from the west, indicating the CLW snowfall travels west to east in this region. This fits with previously mentioned work that showed snowfall in the central GrIS does not come from the north (Shupe et al., 2013; Castellani et al., 2015; P18).

The western GrIS is defined in this work as basins 6.1, 6.2, 7.1, 7.2, 8.1, and 8.2. Its composite (Fig. 16) includes 999 IC cases and 372 CLW cases. Again the mean 500 mb GPH for IC shows a trough to the west of Greenland (Fig. 16a) and the mean for CLW is relatively flat (Fig. 16b). The GPH anomalies for the IC cases again show a dipole, but in this case the high anomaly is now off the southeastern Greenland coastline and the low anomaly is west-northwest of Greenland. The location of the low is possibly suggestive of the mean cyclone found by Chen et al. (1997) located in Baffin Bay and which they connected with increased snowfall in the central west GrIS. The western GrIS CLW events are associated with the same relatively flat 500mb mean GPH and high anomaly over the GrIS (Fig. 16, b and d) seen in the northern and near-Summit composites. The anomalous high ridge and winds show on-shore flow in the central west GrIS. In their work using reanalyses to look at Greenland blocking, Hanna et al. (2016) showed similar 500 mb GPH and wind speed anomaly plots to be associated with positive precipitation anomalies along the western coastline of Greenland.

Moving finally to the southeastern GrIS, defined as basins 3.3, 4.1, 4.2, 4.3, and 5.0, the composites shown in Fig. 17 are made up of 422 IC cases and 114 CLW cases. For this region the IC mean 500 mb GPH (Fig. 17a) shows the deepest trough of the four regions, similarly placed to the west of Greenland but now extending all the way to the southern tip. This is consistent with previous studies connecting north Atlantic stormtracks to heavy precipitation in the southeastern GrIS (e.g. Chen et al., 1997; Schuenemann et al., 2009; Vihma et al., 2016; Berdahl et al., 2018). The IC anomaly plot (Fig. 17c) has a strong dipole with the trough just to the west of the southern tip of the GrIS and the ridge centered just east of Iceland. The anomalous winds associated with the dipole are high and flowing from the North Atlantic onto the southeastern GrIS. This scenario is suggestive of lee cyclogenesis, where cyclones form in the lee of the topographic ridge running along the southern tip of Greenland

(Rogers et al., 2004; Schuenemann et al., 2009). Lee cyclogenesis has been found previously to correlate with precipitation in the southern region (Chen et al., 1997; Schuenemann et al., 2009). Indeed, Berdahl et al. (2018) found that the position of the Icelandic low is a determining factor in the amount of snowfall hitting the southeastern coastline, with up to a 40 % increase when the low is in its far west position, relative to its far east position. The CLW GPH mean (Fig. 17b) has a very shallow trough-ridge feature, in the same location but much less distinct than for the IC events. The CLW anomaly (Fig. 17d) shows a dipole, but the low is now fully off the continent, and most of the GrIS is under an anomalous high that is directing onshore flow from the North Atlantic into the southeastern GrIS.

In all of these regional composites, there are common themes for both the IC and the CLW cases. For each region the two regimes are related to clearly distinct circulation patterns. The IC anomalies tend to have a trough and ridge dipole centered around the particular region of interest with anomalous flow directed onshore into the region in a pattern that resembles previously identified cyclone activity. The 500 mb GPH anomalies for the CLW consistently show an anomalous ridge over the GrIS, but centered in such a way that the anomalous winds are directing flow and moisture into the region from the nearest coastline. These results point to IC snowfall being consistently associated with cyclone activity while CLW snowfall occurs under quieter, high pressure scenerios that favor long lived Arctic mixed-phase clouds. Understanding how circulation patterns relate to snowfall may allow for improvements in our predictions of future changes to GrIS snowfall. For example, climate models predict significant changes in both the paths and intensities of North Atlantic cyclones in response to climate change (Zappa et al., 2013), which will undoubtably impact IC snowfall frequency and accumulation over the GrIS.

## 5   Conclusions

Motivated by the results in P18, we used CloudSat and CALIPSO observations to quantify the frequency and rate of snowfall over the Greenland Ice Sheet (GrIS) associated with two distinct microphysical regimes: snowfall produced by fully glaciated clouds (IC) and snowfall produced by Arctic mixed-phase clouds containing super-cooled liquid water (CLW).

IC snowfall is responsible for $\sim$80 % of the 399 Gt yr$^{-1}$ estimated annual snowfall accumulation over the GrIS from January 2007 to August 2016, the remainder derives from CLW snowfall. IC also dominates the annual snowfall frequency, making up $\sim$70 % of observed events. The relative contributions from the two regimes exhibit pronounced seasonal cycles in both rate and frequency. Monthly snowfall accumulation in summer (May - September) is higher than in winter (October - April). Summer also experiences the peak in CLW frequency, with $\sim$45 % of all GrIS snowfall observations associated with liquid containing clouds during this season. The vast majority of winter season snowfall originates from IC events ($\sim$84 %), and, while the mean winter snowfall rates are lower, the highest individual snowfall rates were observed in winter.

Annual GrIS snowfall frequency exhibits both a strong north-south gradient with snowfall occurring more frequently on the southern portion of the ice sheet and an east-west gradient in regime frequency, with CLW making up a larger fraction of events along the western GrIS than the eastern side. The north-south gradient is enhanced in winter months, with the highest frequency along the steep orography of the GrIS's southeastern coastline. In this region and season, total snowfall frequency decreases with increasing elevation, the result of a combination of a steady decrease in CLW events with increasing elevation

while IC events increase to a maximum between 1,000–1,500 m and decrease above that level. By contrast, in summer the total snowfall frequency increases with elevation, CLW events maintain an almost constant frequency while IC increase with elevation. The snowfall rate for both regimes decreases with increasing elevation in winter and summer, and summer rates are generally higher than winter rates at each level.

IC events in all regions appear to be associated with cyclones interacting with the GrIS, while CLW events are coincident with anomalously high 500 mb geopotential heights over the GrIS. In both regimes, anomalous winds direct flow and moisture onto the GrIS from the nearest coastline. The wintertime north-south gradient in frequency thus arises from North Atlantic cyclones interacting with the steep southeastern coastline, producing IC dominated snowfall as the air masses come onshore. The east-west gradient in CLW relative frequency could result from the slow southwest and westerly mean winds and large-

scale anomalous high pressure, which give rise to widespread conditions favorable to the formation of Arctic mixed-phase clouds and that could encourage their west-to-east propagation.

Mean geometric cloud depth for all GrIS observed snowfall is ∼4 km for IC events and ∼2 km for CLW events. Cloud depth decreases with elevation for both regimes. The thickest snowing clouds that occur on the GrIS are summer IC events in the southeastern GrIS.

Comparisons with ground-based observations showed that CloudSat's CPR is sensitive enough to detect most IC events (∼95 %) as defined at the surface. However the CPR struggles somewhat with the shallower CLW events, identifying a smaller fraction of those defined at the surface (∼75 %). This likely results in an overall underestimation of GrIS snowfall accumulation by as much as 10 % and, in particular, the component owing to CLW events. While a future satellite based measurement system may reduce the depth of the blind zone, these missed cases represent a persistent limitation for CloudSat's CPR.

The CloudSat total GrIS snowfall estimate of 399 Gt yr$^{-1}$ is lower than previously published estimates (e.g. Cullather et al., 2014), however there is evidence that GrIS snowfall has been overestimated relative to surface observations in both reanalyses (Bromwich et al., 2016; Koyama and Stroeve, 2019) and models (Berdahl et al., 2018).

Our results, derived from a decade of satellite observations between 2007 and 2016, provide a snapshot of snowfall characteristics in a rapidly changing Arctic. This snapshot provides insights into the character of present-day snow events across the

GrIS and the dominant synoptic patterns that produce them. The large scale atmospheric circulation in the Arctic is predicted to change with global warming (e.g. Zappa et al., 2013). Combining our regime based results with climate model predictions of future circulation patterns may yield additional insights into how this important source of ice sheet mass may change in a warmer climate.

*Data availability.*   The satellite derived data products used in this study (2C-SNOW-PROFILE, 2B-CLDCLASS-LIDAR, and 2B-GEOPROF)

are publicly available from the CloudSat Data Processing Center: http://www.cloudsat.cira.colostate.edu/data-products

The ground-based snow classification product used in this study is publicly available from the National Science Foundation Arctic Data Center: https://doi.org/10.18739/A2R28Q (Pettersen and Merrelli, 2018)

The MMCR data used in this study are publicly available from the National Science Foundation Arctic Data Center: https://doi.org/10.18739/A20G3GZ8B (Shupe, 2010)

The reanalysis data used in this study are available from ECMWF: https://confluence.ecmwf.int/display/CKB/ERA5+data+documentation. Note: Neither the European Commission nor ECMWF is responsible for any use that may be made of the Copernicus Information or Data it contains.

## Appendix A: List of acronyms

A list of all acronyms used in this manuscript.

**2BCCL**       **2B-**C**LD**C**LASS-**L**IDAR** (cloud dataproduct)

**2BG**       **2B-**G**EOPROF** (reflectivity dataproduct)

**2CSP**       **2C-**S**NOW-**P**ROFILE** (snowfall dataproduct)

**AGL**       **A**bove **G**round **L**evel

**CALIOP**       **C**loud–**A**erosol **Li**dar with **O**rthogonal **P**olarization (aboard CALIPSO)

**CALIPSO**       **C**loud-**A**erosol **L**idar and **I**nfrared **P**athfinder **S**atellite **O**bservation

**CLW**       **C**loud containing **L**iquid **W**ater (refering to snowfall regime)

**CPR**       **C**loud **P**rofiling **R**adar (aboard CloudSat)

**ECMWF**       **E**uropean **C**entre for **M**edium-Range **W**eather **F**orecasts

**GPH**       **G**eo**p**otential **H**eight

**GrIS**       **Gr**eenland **I**ce **S**heet

**ICECAPS**       **I**ntegrated **C**haracterization of **E**nergy, **C**louds **A**tmospheric State
and **P**recipitation at **S**ummit

**IC**       **I**ce **C**loud (refering to snowfall regime)

**MMCR**       **M**illi**m**eter **W**avelength **C**loud **R**adar

**MWR**       **M**icro**w**ave **R**adiometer

**P18**       **P**ettersen et al. (20**18**)

**POSS**       **P**recipitation **O**ccurrence **S**ensor **S**ystem

*Author contributions.* E. A. McIlhattan was responsible for the overall conceptualization and methodology of this work. She led the investigation, conducted the formal analysis of the satellite data, wrote the original draft, completed revisions based on co-author review, and completed the data visualization for Figures 1 - 13. C. Pettersen assisted with the conceptualization of the work, gave pre-publication critical review, and developed the methodology and completed the visualization for the reanalysis based atmospheric circulation patterns shown in Figures 14, 15, 16, and 17. N. B. Wood gave pre-publication critical review and developed the methodology and conducted the formal analysis for the evaluation of MMCR data to determine missed-cases over Summit Station, producing the data in Table 1. T. S. L'Ecuyer provided computing resources, pre-publication critical review, and assistance with the overall conceptualization and methodology for this work.

*Competing interests.* The authors declare that they have no conflict of interest.

*Acknowledgements.* The work by E. A. McIlhattan was funded by a National Aeronautics and Space Administration Earth System Science Fellowship (NNX16AN99H). C. Pettersen's contribution was supported by National Science Foundation awards (1304544, 1801318, 1355654). N. B. Wood's contribution was performed at the University of Wisconsin - Madison for the Jet Propulsion Laboratory, sponsored by the National Aeronautics and Space Administration. T. S. L'Ecuyer's contribution was supported by a CloudSat/CALIPSO science team grant (NNX16AP186).

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

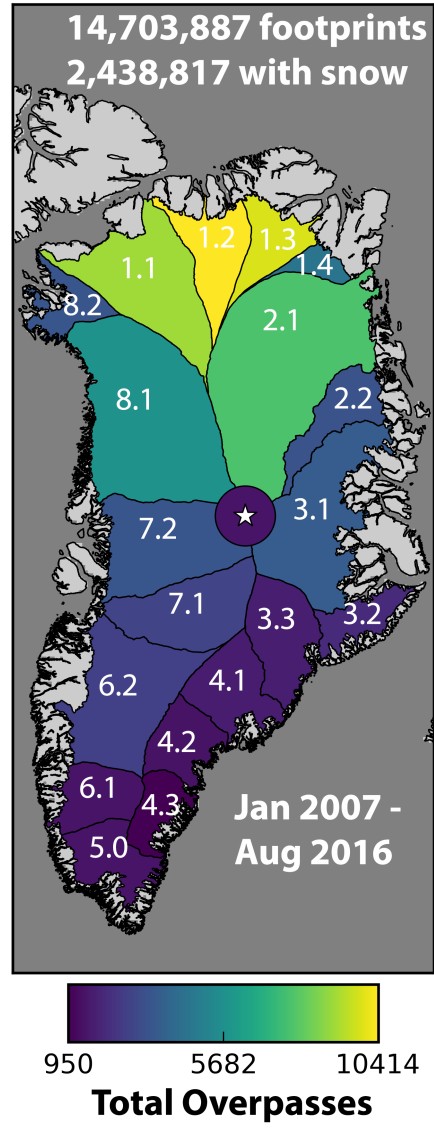

**Figure 1.** A summary of the CloudSat/CALIPSO satellite observations collected over the Greenland Ice Sheet (GrIS). The GrIS is divided into the drainage basins defined by the Ice Altimetry group at Goddard Space Flight Center (Zwally et al., 2012). The color scale represents the total number of satellite overpasses in each basin during the full study period, January 2007 through August 2016. During that period, there were 14,703,887 individual satellite observations, 2,438,817 of which contained snowfall. The star indicates the location of Summit Station, and the circle is the 100 km radius surrounding Summit.

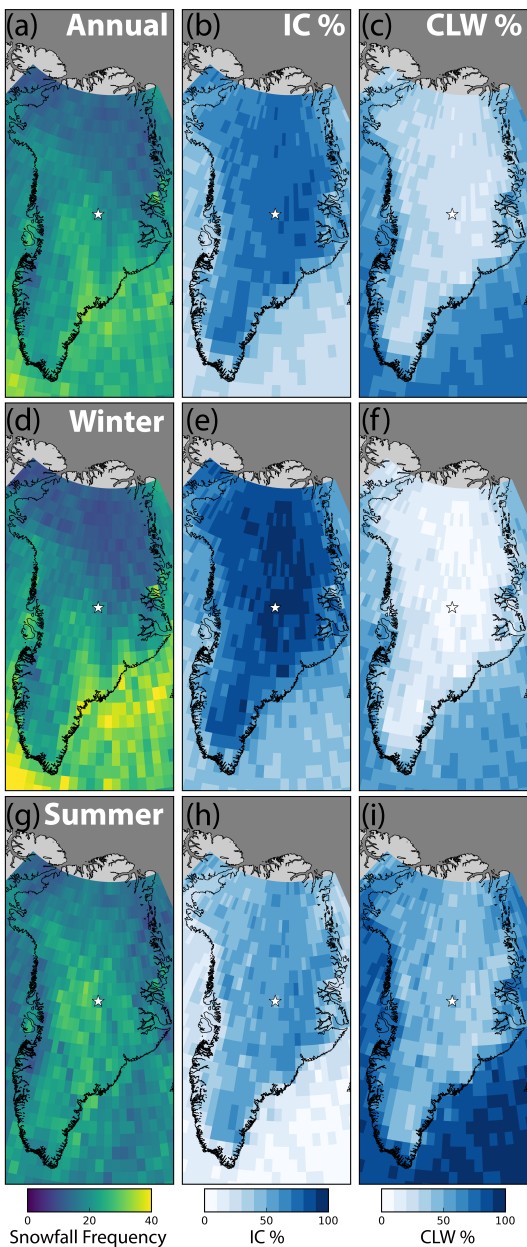

**Figure 2.** Snowfall frequency over the GrIS defined as observations of snowfall divided by total observations in each gridbox. **(a)** is annual mean snowfall frequency using all observations from the study period, **(b)** is the percentage of total snowfall observations that were coincident with ice phase clouds, and **(c)** is the percentage of the total snowfall observations that were coincident with clouds containing liquid water. **(b)** and **(c)** sum to 100. **(d)** is winter mean snowfall frequency (Oct-Apr), with **(e)** and **(f)** the percentages of winter snowfall coincident with ice phase clouds and clouds containing liquid water, respectively. **(g)** is summer mean snowfall frequency (May-Sep), with **(e)** and **(f)** the percentages of summer snowfall coincident with ice phase clouds and clouds containing liquid water, respectively. The location of Summit Station is marked in each panel by a white star.

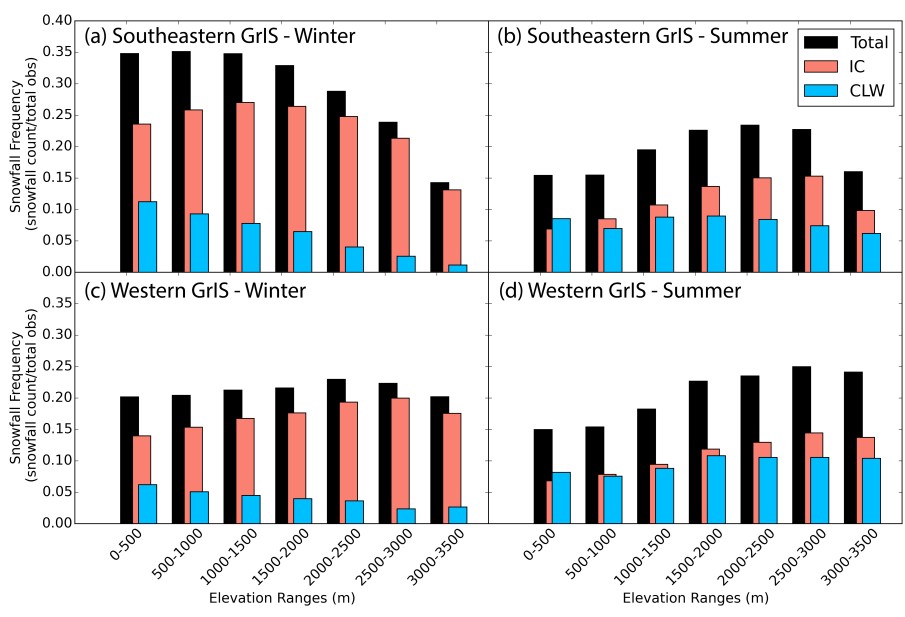

**Figure 3.** Snowfall frequency partitioned by elevation for **(a)** winter (Oct–Apr) in the southeastern GrIS (basins 3.3, 4.1, 4.2, 4.3, and 5.0 in Fig. 1), **(b)** summer (May–Sep) in the southeastern GrIS, **(c)** winter in the western GrIS (basins 6.1, 6.2, 7.1, 7.2, 8.1, and 8.2), and **(d)** summer in the western GrIS. Frequency is determined by dividing the number of snowfall observations by the total number of observations for a given elevation range. Black bars include all observed snowfall, red bars include IC events only, blue bars include CLW events only.

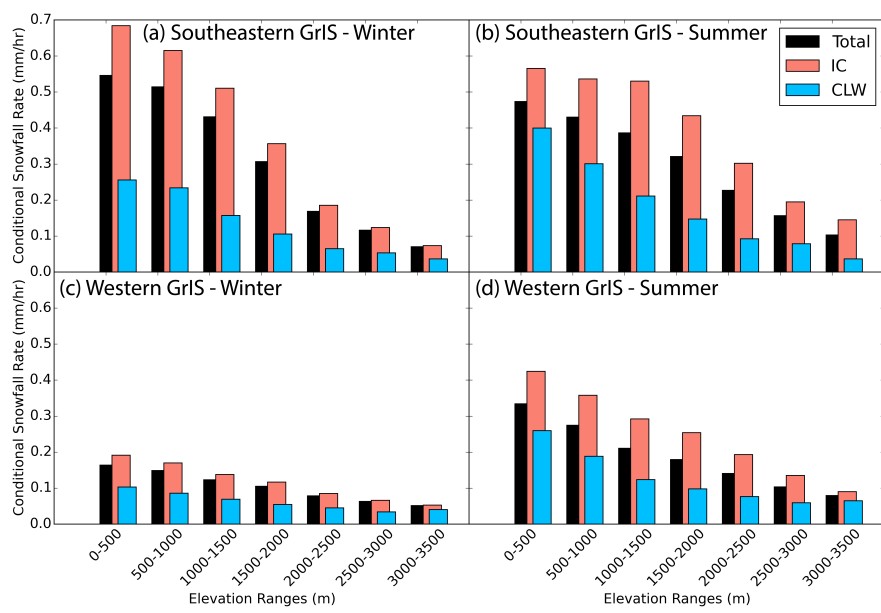

**Figure 4.** As in Fig. 3 with conditional snowfall rate. This is the mean rate of snowfall for all snowfall observations of a given regime.

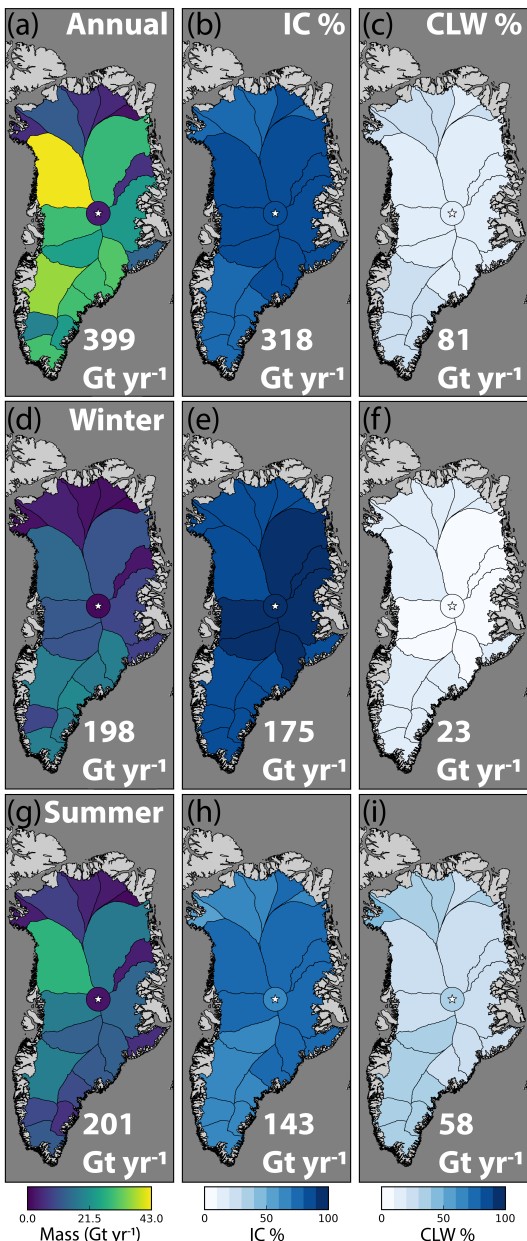

**Figure 5.** Snowfall mass contribution to the GrIS. **(a)** is the annual average mass contribution broken down by basin, with the color scale representing Gt yr$^{-1}$ for each basin and the total mass listed in the bottom right corner. **(b)** is the percentage of the snowfall mass produced by ice clouds, and **(c)** is the percentage of the mass produced by liquid containing clouds. The center (**(d)**, **(e)**, and **(f)**) and bottom (**(g)**, **(h)**, and **(i)**) rows are as the top row but for winter (Oct-Apr) and summer (May-Sep) months, respectively. The location of Summit Station is marked in each panel by a white star and the color of the circle surrounding it indicates the mass/percentage value for the area within 100km radius of the station.

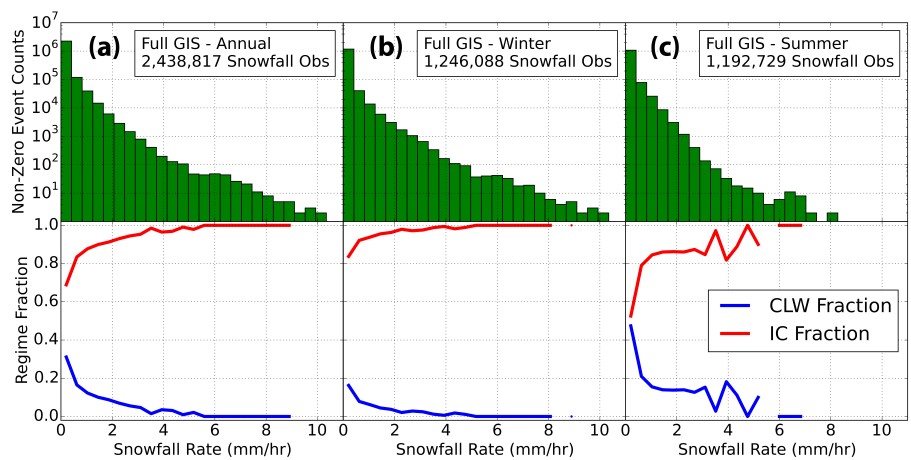

**Figure 6.** Snowfall rates for all observed snowfall. **(a)-top** is a histogram of the observed rates of all GrIS snowfall from 2CSP (log scale), and **(a)-bottom** is the regime percent for each histogram bin. **(b)** and **(c)** are the same for GrIS winter (Oct-Apr) and summer (May-Sep) months, respectively.

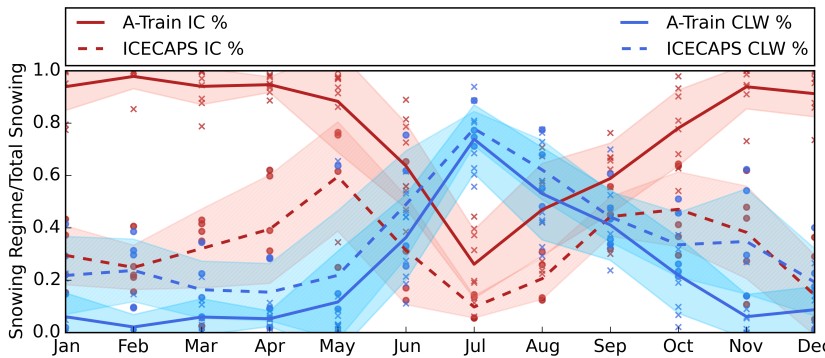

**Figure 7.** Annual cycle of regime fraction near Summit Station, Greenland. The regime fraction is the number of observations of one of the snowfall regimes (IC or CLW) divided by the total number of snowfall observations. A-Train values (solid lines, 'x' markers) shown for the near Summit annual cycle line plot are averages for all CPR footprints within 100km of Summit Station, Greenland. The solid lines represent the average of all observational years, each x depicting a single year's monthly average. The shaded region surrounding each line is the standard deviation about the mean for the month. The red color is for the IC regime percent and the blue is for the CLW regime percent. For the A-Train data, the red and blue add to 1.0. The ICECAPS values (dashed lines, circle markers) are from vertically pointing instruments at Summit Station, also with markers representing a single year's monthly average and the line being the mean of all years. The ICECAPS IC and CLW data do not add to 1.0 because of an additional category of 'indeterminate' (not plotted).

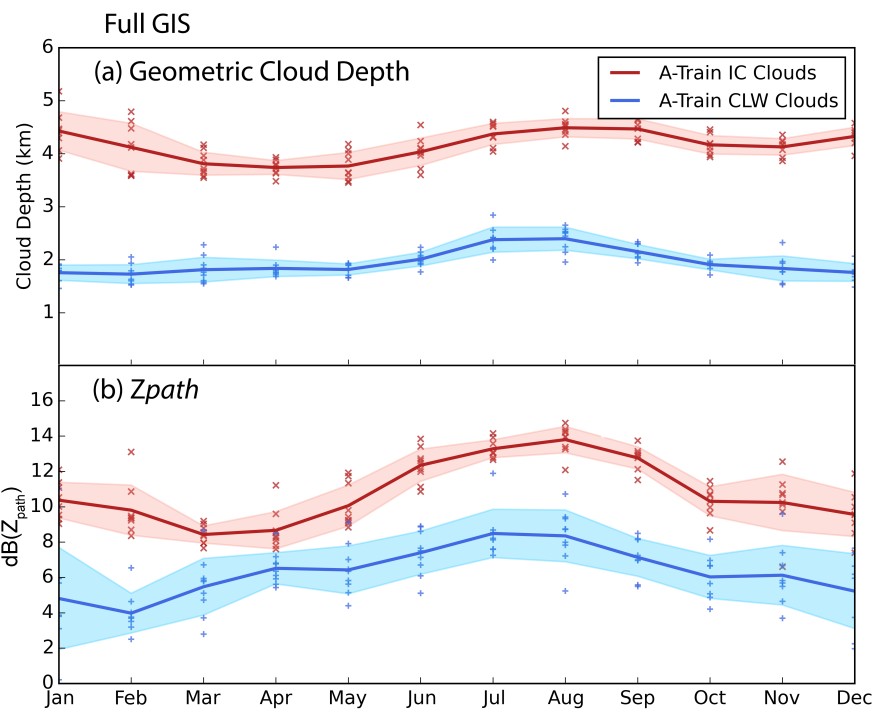

**Figure 8.** Annual cycle of GrIS snowfall cloud characteristics. **(a)** the geometric cloud depth, and **(b)** the vertically integrated reflectivity for IC (red) and CLW (blue) snowfall observations. The solid lines represent the average of all observational years, each marker (x,+) depicting a single year's monthly average. The shaded region surrounding each line is the standard deviation about the mean for the month.

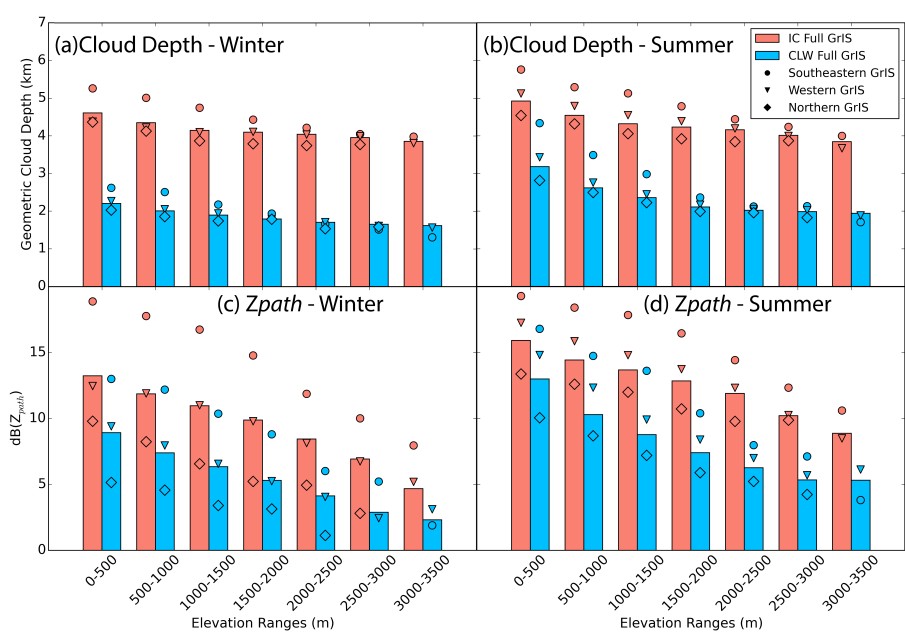

**Figure 9.** Geometric cloud depth partitioned by elevation for **(a)** winter (Oct-Apr) and for **(b)** summer (May-Sep). Column integrated CPR reflectivity within the cloud layer ($dB(Z_{path})$) for **(c)** winter and **(d)** summer. The red bars are the mean value at each elevation for all IC events across the full GrIS for the given season and elevation. The blue bars are the means for the CLW events. The symbols represent the means for three GrIS regions: the southeastern GrIS (filled circles, basins 3.3, 4.1, 4.2, 4.3, and 5.0 in Fig. 1), the western GrIS (filled triangles, basins 6.1, 6.2, 7.1, 7.2, 8.1, and 8.2), and the northern GrIS (filled diamonds, basins 1.1, 1.2, 1.3, and 1.4).

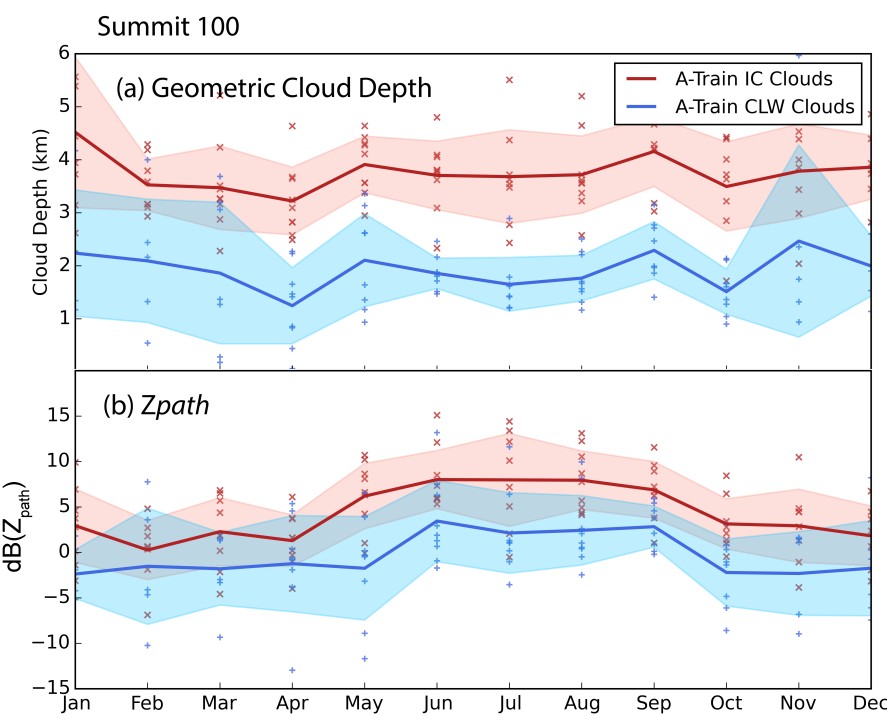

**Figure 10.** As in Fig. 8 except only including observations within a 100 km radius of Summit Station.

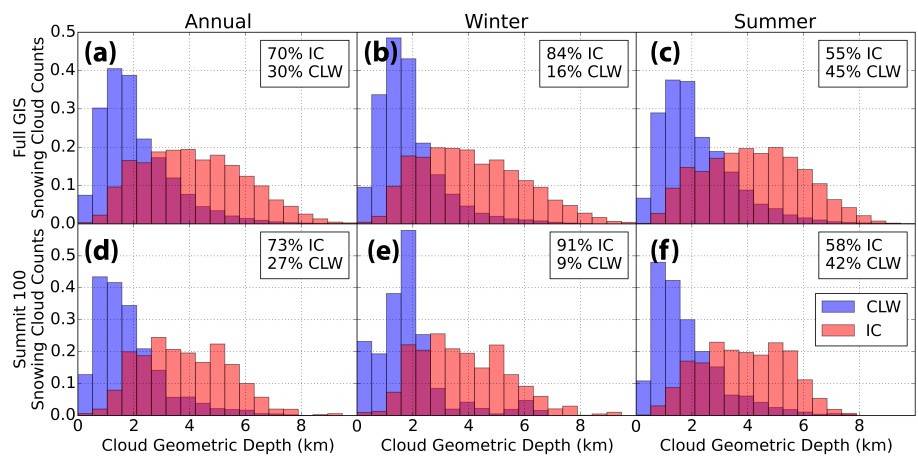

**Figure 11.** Histograms of precipitation regime geometric cloud depth. Red bins contain all footprints of IC snowfall and blue bins contain all footprints of CLW snowfall for each given season (Annual, Winter, and Summer) and region (Full GrIS, Summit 100) as described in Fig. 6. The histograms are normalized to highlight the distribution differences. The relative percentage of each regime is listed in the top right of each panel.

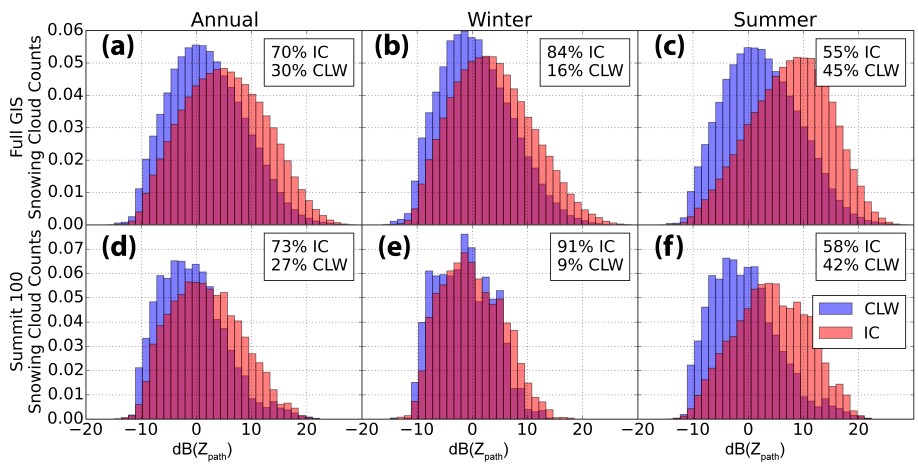

**Figure 12.** As in Fig. 11 with $dB(Z_{path})$.

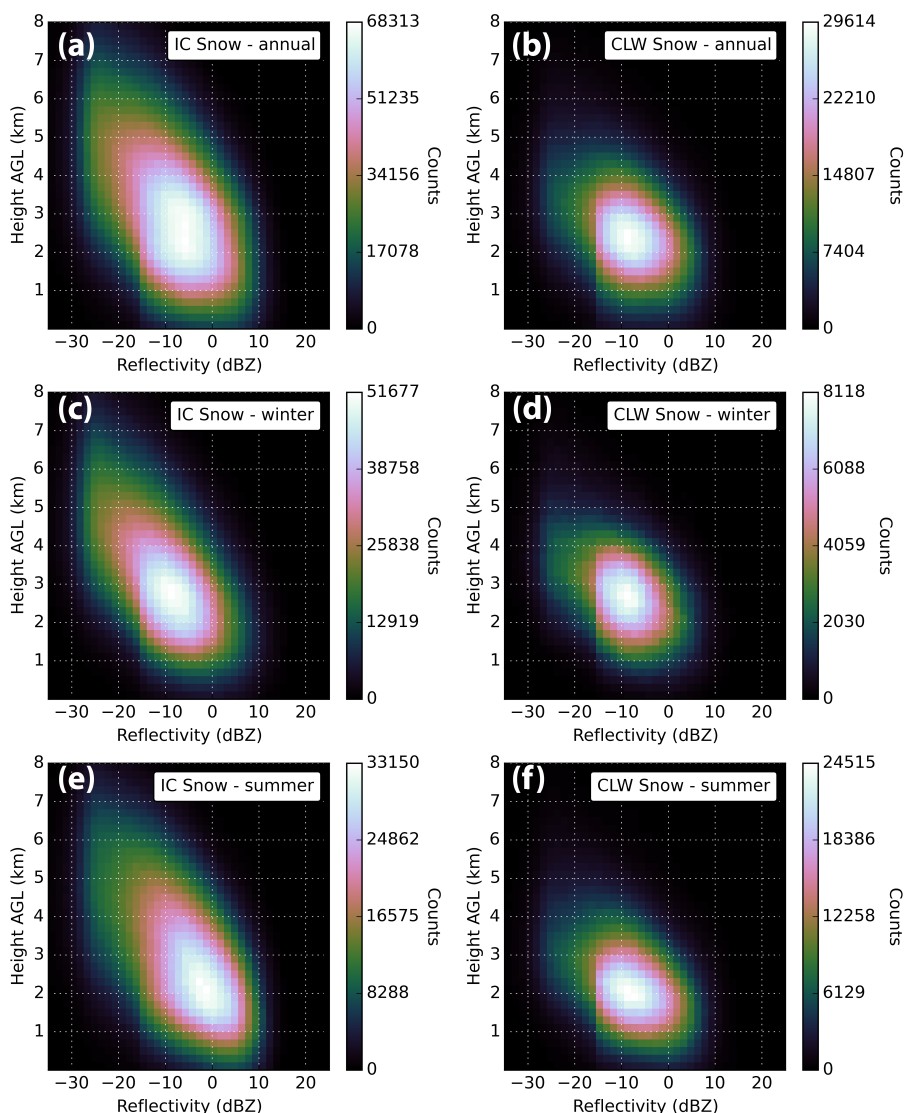

**Figure 13.** Composite two-dimensional histograms of CPR heights and reflectivities for the two snowfall regimes over the full GrIS. The top row contains the entire annual cycle of events, including every footprint of snowfall detected during the study period, for IC **a)** and CLW **(b)** events. The center row contains all wintertime (Oct-Apr) IC **(c)** and CLW **(d)** events, and the bottom row contains all summertime (May-Sep) IC **(e)** and CLW **(f)** events. There is a discontinuity apparent in each panel at ∼-15 $dBZe$. This is due to the 2CSP threshold of -15 $dBZe$ for defining snowfall events. The shape and character of these plots compare well to P18 Fig. 6.

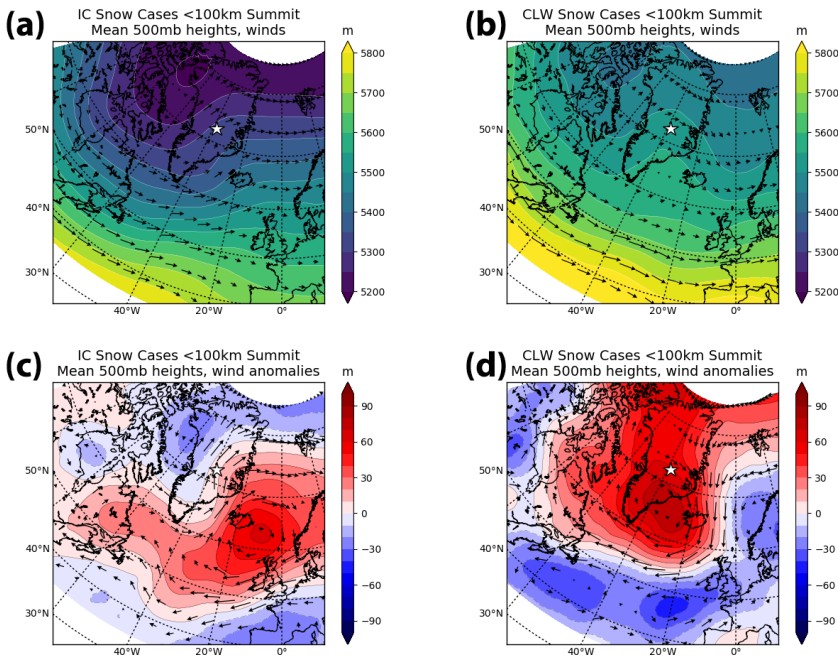

**Figure 14.** ERA5 derived mean and anomaly 500 mb geopotential heights (GPH) and winds for the strongest 50 % of precipitation events that occurred within a 100 km radius of Summit Station during the study period. **(a)** shows the average 500 mb GPH and winds for 159 IC events and **(b)** shows the same for 43 CLW events. **(c)** and **(d)** show the GPH and wind anomalies for the IC and CLW cases, respectively. These panels are all consistent with P18 Fig. 11, which also shows a strong trough ridge for the IC snow cases and relatively calm, quiescent conditions for the CLW snow cases.

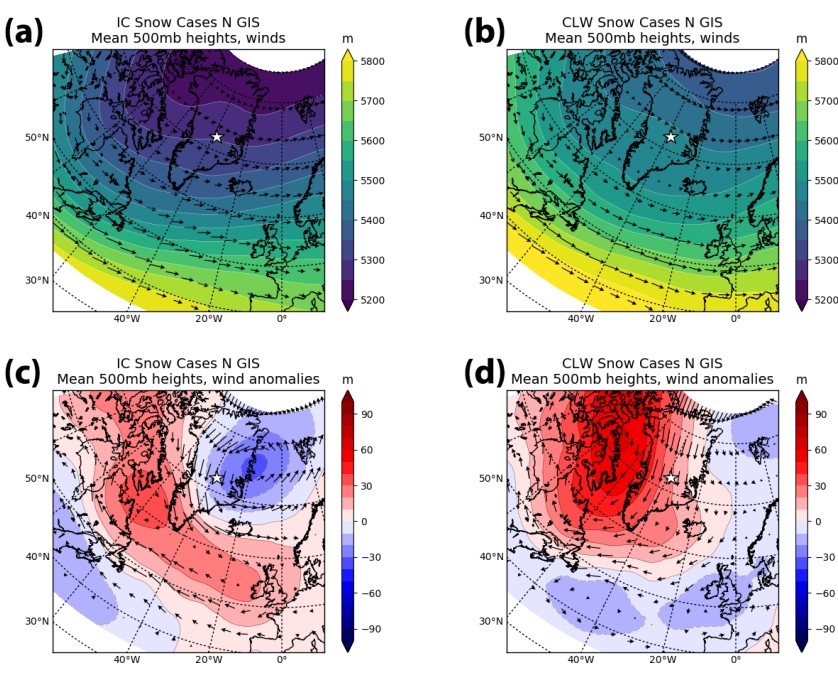

**Figure 15.** As in Fig. 14 for the northern GrIS: basins 1.1, 1.2, 1.3, and 1.4.

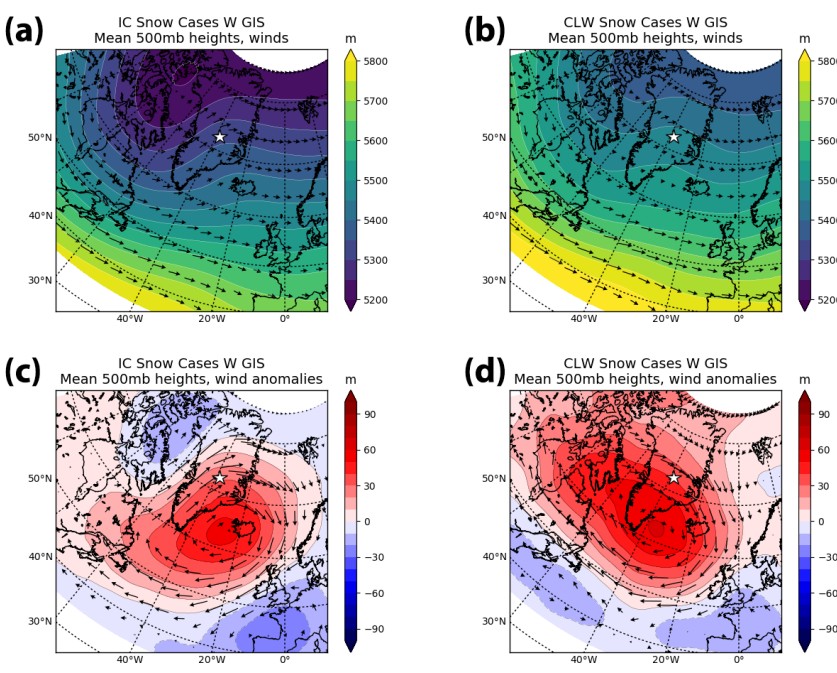

**Figure 16.** As in Fig. 14 for the western GrIS: basins 6.1, 6.2, 7.1, 7.2, 8.1, and 8.2.

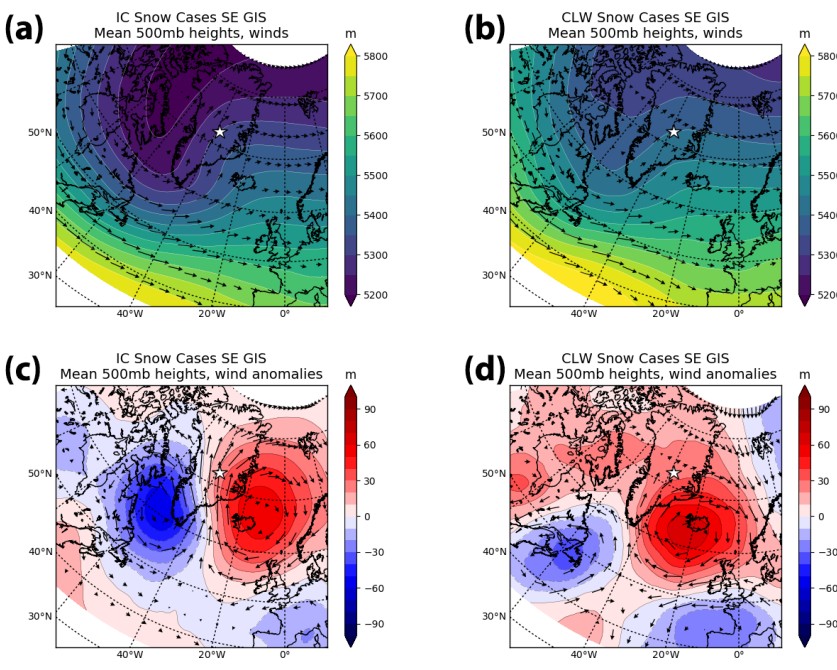

**Figure 17.** As in Fig. 14 for the southeastern GrIS: basins 3.3, 4.1, 4.2, 4.3, and 5.0.

**Table 1.** Summary of CloudSat snowfall detection capability over Summit Station, Greenland based on averaged MMCR data for POSS detected snowfall. To mimic CloudSat detection we used: a height range of 960-1200 m, equivalent to the standard height of bin 5 of the CPR used in the 2CSP algorithm over land; and time average of 300 s, which at a moderate wind speed of 5 m s$^{-1}$ is equivalent to the horizontal CPR footprint of ~1.5 km.

| Snowfall Event Type | # of Events | Total Fraction Missed | Mean Rate - Missed | Mean Rate - Detected |
|---|---|---|---|---|
| Total Snow Events | 20,516 | 0.22 | 0.05 | 0.09 |
| IC Only Events | 3,545 | 0.05 | 0.05 | 0.10 |
| CLW Only Events | 9,777 | 0.25 | 0.05 | 0.10 |

Analysis of CPR-like MMCR reflectivities.

**Table 2.** Summary of 2CSP accumulation estimates by GrIS basin. All masses are in Gt yr$^{-1}$. The "summit100' basin includes every observation within 100km of Summit Station.

| Basin # | Annual Mass (IC%,CLW%) | Winter Mass (IC%,CLW%) | Summer Mass (IC%,CLW%) | Area km$^2$ |
|---|---|---|---|---|
| 1.1 | 12 ( 77 , 23 ) | 4 ( 88 , 11 ) | 8 ( 62 , 38 ) | 131,115 |
| 1.2 | 7 ( 78 , 22 ) | 2 ( 87 , 13 ) | 4 ( 67 , 33 ) | 63,773 |
| 1.3 | 5 ( 83 , 17 ) | 2 ( 90 , 10 ) | 3 ( 74 , 26 ) | 46,152 |
| 1.4 | 2 ( 79 , 21 ) | 1 ( 87 , 13 ) | 1 ( 67 , 33 ) | 17,536 |
| 2.1 | 29 ( 82 , 18 ) | 11 ( 91 , 9 ) | 17 ( 71 , 29 ) | 274,220 |
| 2.2 | 6 ( 85 , 15 ) | 3 ( 91 , 9 ) | 4 ( 76 , 24 ) | 51,196 |
| 3.1 | 23 ( 87 , 13 ) | 9 ( 94 , 6 ) | 14 ( 78 , 22 ) | 148,090 |
| 3.2 | 14 ( 81 , 19 ) | 8 ( 89 , 11 ) | 6 ( 70 , 30 ) | 35,619 |
| 3.3 | 31 ( 86 , 14 ) | 18 ( 92 , 8 ) | 13 ( 78 , 22 ) | 73,232 |
| 4.1 | 30 ( 84 , 16 ) | 18 ( 90 , 10 ) | 12 ( 76 , 24 ) | 64,669 |
| 4.2 | 30 ( 77 , 23 ) | 20 ( 85 , 15 ) | 10 ( 67 , 33 ) | 46,802 |
| 4.3 | 24 ( 73 , 27 ) | 18 ( 80 , 20 ) | 6 ( 64 , 36 ) | 33,326 |
| 5.0 | 30 ( 77 , 23 ) | 18 ( 85 , 15 ) | 12 ( 64 , 36 ) | 49,738 |
| 6.1 | 19 ( 79 , 21 ) | 9 ( 87 , 13 ) | 10 ( 67 , 33 ) | 49,909 |
| 6.2 | 36 ( 78 , 22 ) | 18 ( 86 , 14 ) | 18 ( 66 , 34 ) | 136,902 |
| 7.1 | 24 ( 83 , 17 ) | 11 ( 92 , 8 ) | 13 ( 70 , 30 ) | 95,213 |
| 7.2 | 30 ( 84 , 16 ) | 12 ( 91 , 9 ) | 18 ( 73 , 27 ) | 130,027 |
| 8.1 | 42 ( 82 , 18 ) | 14 ( 90 , 10 ) | 28 ( 71 , 29 ) | 241,556 |
| 8.2 | 6 ( 74 , 26 ) | 2 ( 85 , 15 ) | 4 ( 59 , 41 ) | 33,497 |
| summit100 | 3 ( 83 , 17 ) | 1 ( 93 , 7 ) | 2 ( 69 , 31 ) | 31,416 |

Summary of snow accumulation by basin.