# Peer review of "Satellite Observations of Snowfall Regimes over the Greenland Ice Sheet"

_The Cryosphere, 2019_

## Referee Comment (RC1) · Anonymous Referee #1 · 27 Dec 2019

major comments...

At the highest level, the article would benefit by being shortened, streamlined to emphasize whatever new it brings, avoid re-stating what is already well known, condense text where possible. I have pointed out some novelties I recommend get more emphasis.

The main value of the study is to "look beyond Summit station", so please 1.) streamline the Summit text 2.) add more discussion emphasis to other regions that goes beyond synoptic climatology (discussion of obvious and already documented precipitation being associated with troughs)... There has to be something about the satellite perspective taht brings much more than synoptic climatology does; for example, why not use a heavy precipitation event case to make some new points?

I don't entirely agree with the statement "Snowfall accumulation is the only significant, positive term in the surface mass balance of the GIS" ... surface water vapor deposition can be, as Box and Steffen (2001) abstract: "At high-elevation sites; the annual water vapor flux is positive, up to +32 ±9 mm at the North Greenland Ice core Project (NGRIP) and +6 ±2 mm at Summit." or above 25 mm near Summit which is roughly 10% of the accumulation rate (see Fig. 6 2 Level method). ... so the issue of the remote sensing technique not observing? water vapor deposition needs some treatment in this study if not just clear recognition water vapor deposition is one of the underestimates of precipitation. similarly, I am not that comfortable with "evaporation over the snow- and ice-covered regions of the Arctic is negligible"... the issue of how much moisture is recycled from daytime sublimation and nighttime deposition deserves at least some mention.

Move discussion text (for example page 17 lines 3-22) out of conclusions section. In conclusion section, limit text to what new this study finds, little more please. How to accomplish is to make a list of new insights in this section, simple as that!

By the way, I think you probably agree, it would be helpful to have all the fancy equipment lower on the ice sheet, for example, near DYE-2 and much closer to the airport! You're welcome to pass my comment to your science foundation.

about my comments... io means instead of comments, if two numbers, first number is page number, second is line number

minor comments...

throughout consider "study" io "paper"

page 1 abstract "due to increasing surface melt" io "due to surface melt"

After it is soon obvious the region is the Greenland ice sheet, use "ice sheet" io "GIS"

11-12 "Overall, most CPR observations of snowfall over the GIS come from IC events (70 %), however, during the summer months, close to half of the snow observed is

produced in CLW events (45 %)." ... really depends where and as the next sentence puts it, when, i.e. "summer", so what is the point?

17 "growth" of ___? be clearer

18 how is "large scale anomalous high pressure" different from "large scale high pressure"? The latter is less ambiguous IMO

18 "Ground-based data" be more specific; location, sensor

22 "key role in both the global energy budget (e.g. Box et al., 2012)" that study is not exemplary of global energy budget and "key" here is vague, pls rephrase

24 "year-round" actually not in winter or at night

page 2

Bamber et al 2013 have a more accurate number than "7.2 m (Church et al., 2001)"

7 "Recent mass loss" add time interval(s)

"shortwave" io "SW"

"summer" is not the right word, non-summer months can matter. More meaningful can be to write of "sunlit periods" or "period of positive net radiation"

21 "surface height" worth having a look at PROMICE.org data and associated publications

24-25 "wide range of GIS snowfall estimates" see work of Lewis in Cryosphere, a paper from 2018? and another now in review 2019 TCD

26 "ground-based snow observations" io "ground-based observations" 26 "can be useful to examine" io "are needed to look at" 27 "from space"... "remotely" o 31 "an attractive " io "currently the best "

6 19 "explore the contents of clouds" rephrase to be more explicit re: "contents" ..."examine" may be better than "explore" ] 8 15 and throughout the paper, I suggest replacing "look" with "examine" ..."examined" io "looked specifically at" 10 30 remove "(âĹij1.2 million)" un-needed and arbitrary 9 16 "The frequency of detected snowfall events" io "The frequency of snowfall" 9 22-23 "more CLW cases along the western side of the GIS than the eastern" a point may be worth emphasizing as a new result this study brings forward. The previous information in the paragraph is already common knowledge in the field 9 25 winter "stronger north-south gradient compared to the annual distribution" also deserving some highlight, perhaps buried here, put it in conclusions if not also in abstract 10 5 "âĹij83 % of its annual incoming solar insolation." according to what info? 10 8 not in full agreement with "Snowfall accumulation is the only significant, positive term in the surface mass balance of the GIS" see major comments 10 34 "hydrometeor" mean rain? be clear or use less jargon 11 1-2 "while the common summer events are snowing at slightly higher rates on average, it is in winter that the less frequent, highest-intensity snowfall occurs." interesting point worth emphasizing in conclusions 11 13 "determine" io "see", modify throughout 13 31-32 "Excluding a minor moisture recycling in the surface boundary layer that delivers frost during net surface radiative cooling events, the moisture required to produce GIS snowfall is not produced locally" io "The moisture required to produce GIS snowfall is not produced locally" ... actually the first sentence in 4.3 can be removed, the following sentence makes the point and there you might as well mention temperature inversion and PBD moisture recycling
* * *

---

## Referee Comment (RC2) · Anonymous Referee #2 · 8 Jan 2020

The manuscript entitled "Satellite Observations of Snowfall Regimes over the Greenland Ice Sheet" by McIlhattan et al. uses CloudSat and CALIPSO data to quantify the snowfall over the entire Greenland Ice Sheet. Considering the different regimes of snowfall, the authors extend a previous work of Pettersen et al. from Summit station to the entire Greenland and show interesting findings that can justify underestimation results from Bennartz et al. from a snowfall type rather than from an algorithm perspective. Moreover seasonal results are presented, analyzing how the different regimes impact the accumulation of snowfall, and discussing the annual cycles of precipitation and how they impact the albedo of GIS. Cloud properties and atmospheric circulations models are also considered helping understanding the dynamics that actually impact the region. The manuscript is overall well written and fits the scope of the journal. Data

and methods are described in detail and all the results are thoroughly described and discussed. There are only a couple of very minor comments that the authors could address:

Page 3 Line 14: "been" is repeated.

Page 12 Line 15: "have" is repeated.

Page 12 Line 30-35: you are talking here about opposite skewness of the distribution between winter and summer, but I don't actually see a negative skew in summer (with the peak to the right and the tail to the left). I would actually still see a tail to the right for summer even if steeper. I'll leave to the authors if considering this very minor comment or not.

Page 13 Line 1: reading "a distinct nature of the two regimes" makes me think that I should be able to separate the two regimes from the histograms in fig. 9 and this actually not the case since the two distributions are quite totally overlapping. I then understand that these plots tell the story of the IWP behavior for the two regimes, but I would try to describe it better not to lead the reader to a wrong conclusion (being able to distinguish between the regimes thanks to the dBZpath).

---

## Author Comment (AC1) · 13 Mar 2020

We would like to thank the two anonymous referees for their time and effort in reviewing and providing feedback on our manuscript.

Before addressing individual comments, we would like to make clear that the general topic of our manuscript is snowfall. While we motivate our study of Greenland snowfall by talking about the importance of the mass balance of the Greenland ice sheet, our datasets, methods, and results are all from the perspective of precipitation.

The reviewer comments are in red and our responses are in black.  Responses to both reviewers are in one document.

Authors' Response to Anonymous Referee #1

*1. (Referee #1 Comment - major) At the highest level, the article would benefit by being shortened, streamlined to emphasize whatever new it brings, avoid re-stating what is already well known, condense text where possible. I have pointed out some novelties I recommend get more emphasis.*
(Authors' Response) Thank you for the comment. The results that you point out in your later comments (listed as xxiv,xxv,xxix under comment 6) have been highlighted in the conclusion section. In addition, the conclusion section has been condensed to focus more clearly on the novel findings of this work.

*2. (Referee #1 Comment - major) The main value of the study is to "look beyond Summit station", so please 1.) streamline the Summit text 2.) add more discussion emphasis to other regions that goes beyond synoptic climatology (discussion of obvious and already documented precipitation being associated with troughs)... There has to be something about the satellite perspective taht brings much more than synoptic climatology does; for example, why not use a heavy precipitation event case to make some new points?*
(Authors' Response) The emphasis of this study is looking at GrIS precipitation from a satellite. To our knowledge, there is no published work using satellite or remote sensing to look at precipitation regimes over Greenland and their connection to large scale atmospheric circulations, other than Pettersen et al., 2018 which looks only at snowfall at Summit Station. We do reference theoretical and modeling studies of GrIS snowfall (e.g. Chen at al. 1997), reanalysis studies (e.g. Schuenemann et al., 2009), and snowfall implied from ice core studies (e.g. Alley et al., 1993; Kapsner et al., 1995), but the point of this paper is to add a remote sensing perspective. Both Lenaerts et al., 2019 and Bennartz et al., 2019 looked at GrIS snowfall using the same sellites we use, however they did not connect their findings to the synoptic patterns or look at snowfall regimes.

While the satellite approach does allow us to look over the full GrIS, we argue that it is important to focus on Summit Station at the beginning of Section 4.3 in order assess the satellite snowfall retrievals and assure that they are physically realistic before moving on the larger region.

We agree with you that it is well documented and accepted that GrIS precipitation is generally associated with troughs, however, our regional analysis of IC snowfall shows that troughs have differing locations/orientations when producing snow over a given part of the ice sheet. The troughs we show align well with the theories presented in the Chen et al. 1997 paper, which lends confidence to both studies since the remote sensing perspective is independent from the modeling/theory. The ridging that we found to be associated with CLW snowfall is important to record in detail as well, since the connection between ridging and precipitation has been previously documented only in Hanna et al., 2016 (reanalysis based) and Pettersen et al., 2018 (looking only at Summit Station). We believe this is the first documentation of the synoptic patterns associated with GrIS snowfall from the remote sensing perspective, however, if the editor or reviewer have suggestions of additional studies that we should include, we are happy to do so.

Looking at an isolated heavy snowfall event would be anomalous, and we are not sure what new points would be added. If there are particular interesting results that the reviewer or editor would be interested in seeing from a case study of heavy snowfall, it could be the basis of a follow on study, however here we would like to keep the focus on the typical circulation patterns that favor GrIS snowfall. Again, the fact that our observational results agree well with previous theoretical and modeling estimates brings confidence to both perspectives.

*3. (Referee Comment #1 - major) I don't entirely agree with the statement "Snowfall accumulation is the only significant, positive term in the surface mass balance of the GIS" ... surface water vapor deposition can be, as Box and Steffen (2001) abstract: "At high-elevation sites; the annual water vapor flux is positive, up to +32 ±9 mm at the North Greenland Ice core Project (NGRIP) and +6 ±2 mm at Summit." or above 25 mm near Summit which is roughly 10% of the accumulation rate (see Fig. 6 2 Level method). ... so the issue of the remote sensing technique not observing? water vapor deposition needs some treatment in this study if not just clear recognition water vapor deposition is one of the underestimates of precipitation. similarly, I am not that comfortable with "evaporation over the snowand ice-covered regions of the Arctic is negligible"... the issue of how much moisture is recycled from daytime sublimation and nighttime deposition deserves at least some mention.*

(Authors' Response) We agree that at high elevation and/or low precipitation locations, water vapor deposition can be an important contribution to surface accumulation. However, our statement *"Snowfall accumulation is the only significant, positive term in the surface mass balance of the GrIS"* refered to the entire GrIS and it is supported by the references we included:

- "Precipitation is the only significant source term for the mass balance of the Greenland ice sheet and smaller ice caps in the Arctic." Jakobson and Vihma, 2010, p2175
- "The mass budget of the ice sheet as a whole is driven by precipitation at the surface of the ice sheet, which is balanced at the surface by ice melt and runoff." Mottram et al 2019, p1407

For clarity and to address vapor deposition, we changed the statement to read "Snowfall accumulation is the **largest** positive term in the surface mass balance of the GrIS" and have added the following modification to the introduction: "While vapor deposition can be locally important, snowfall is the major source term for the mass of the GrIS…" The satellite instruments that we use don't see the surface, so observing/quantifying the moisture recycling you refer to would be outside the capabilities of our technique.

*4. (Referee #1 Comment – major) Move discussion text (for example page 17 lines 3-22) out of conclusions section. In conclusion section, limit text to what new this study finds, little more please. How to accomplish is to make a list of new insights in this section, simple as that!*
(Authors' Response) Thank you for the suggestion. We have modified the conclusion to focus on the new insights from our work, including the additional results you suggested that we highlight. We have removed the majority of the discussion text out of this section, leaving only three sentences at the end to provide context to the readers on the broader importance of our results.

*5. (Referee #1 Comment) By the way, I think you probably agree, it would be helpful to have all the fancy equipment lower on the ice sheet, for example, near DYE-2 and much closer to the airport! You're welcome to pass my comment to your science foundation.*
(Authors' Response) The instrument suite at Summit Station is funded by the US National Science Foundation and will run through 2020. We are unsure of the future location of the instruments, but will pass along your suggestion.

*6. (Referee #1 Comment – minor revisions) about my comments... io means instead of comments, if two numbers, first number is page number, second is line number*
  i.    *throughout consider "study" io "paper"*
        (Authors' Response) We have made this change.

  ii.   *page 1 abstract "due to increasing surface melt" io "due to surface melt"*
        (Authors' Response) We have made this change.

  iii.  *After it is soon obvious the region is the Greenland ice sheet, use "ice sheet" io "GIS"*
        (Authors' Response) We have updated GIS to GrIS throughout the paper. GrIS avoids the confusion of the previous acronym and is becoming the standard among the atmospheric science community.

  iv.   *11-12 "Overall, most CPR observations of snowfall over the GIS come from IC events (70 %), however, during the summer months, close to half of the snow observed is produced in CLW events (45 %)." ... really depends where and as the next sentence puts it, when, i.e. "summer", so what is the point?*
        (Authors' Response) To our knowledge, this is the first study to partition GrIS snowfall events based on cloud phase, so these numbers are new to the community. We go into more detail in the analysis section regarding the importance of these numbers, the main

idea being that while CLW events only make up 30% of the total GrIS snowfall observations, they make up nearly half (45%) in summer and thus are important for increasing surface brightness during the months where solar radiation is present.

v.    *17 "growth" of ____? be clearer*
(Authors' Response) Since "growth" covers multiple processes that would clutter the abstract (but are detailed in Section 3, page 8, lines 13-17), we have simplified the introduction to say "IC events demonstrate consistently increasing reflectivity toward the surface" rather than "growth toward the surface".

vi.   *18 how is "large scale anomalous high pressure" different from "large scale high pressure"? The latter is less ambiguous IMO*
(Authors' Response) Anomalous high pressure indicates a deviation from the mean in a given region, rather than the actual pressure level. To reduce confusion, we have updated the text to say: "…CLW events generally occur under large scale anomalously high geopotential heights over the GrIS."

vii.  *18 "Ground-based data" be more specific; location, sensor*
(Authors' Response) We have updated the abstract text to say "Ground-based data from an instrument suite at Summit Station is used to estimate…"

viii. *22 "key role in both the global energy budget (e.g. Box et al., 2012)" that study is not exemplary of global energy budget and "key" here is vague, pls rephrase*
(Authors' Response) Thank you for the comment. We have updated that sentence for clarity and replaced the Box et al., 2012 study with Flanner et al., 2011: "The Greenland Ice Sheet (GrIS) is important globally because of its influence on both the energy budget (e.g. Flanner et al., 2011) and water cycle (e.g. Church et al., 2001, Enderlin et al., 2014)."

ix.   *24 "year-round" actually not in winter or at night*
(Authors' Response) We have removed "year-round".

x.    *page 2 Bamber et al 2013 have a more accurate number than "7.2 m (Church et al., 2001)"*
(Authors' Response) Thank you for pointing out this reference, we have updated the number and citation accordingly.

xi.   *7 "Recent mass loss" add time interval(s)*
(Authors' Response) We have added the time interval specific to the referenced study: "Between 1972 and 2018, the GrIS contributed 13.7 mm to global sea level rise…"

xii.  *"shortwave" io "SW"*
(Authors' Response) We have removed the abbreviation SW.

*xiii.* *"summer" is not the right word, non-summer months can matter. More meaningful can be to write of "sunlit periods" or "period of positive net radiation"*
(Authors' Response) Thank you for the suggestion, we have updated the text to say "sunlit periods".

*xiv.* *21 "surface height" worth having a look at PROMICE.org data and associated publications*
(Authors' Response) Thank you for the suggestion, we have looked at the PROMICE website/publications and included a sentence on ground penetrating radar accumulation estimates in the introduction: "Both airborne and ground-based radars have looked below the surface of the GrIS to provide historical accumulation values (Miege et al., 2013; Lewis et al., 2017), but are limited by the specific location of the transects, complications from melt events, and accumulation estimates are for annual or longer periods."

*xv.* *24-25 "wide range of GIS snowfall estimates" see work of Lewis in Cryosphere, a paper from 2018? and another now in review 2019 TCD*
(Authors' Response) In looking at the two Lewis papers in the Cryosphere ("Regional Greenland accumulation variability from Operation IceBridge airborne accumulation radar," 2017; and "Recent precipitation decrease across the western Greenland ice sheet percolation zone," 2019), they provide regional estimates using airborne and ground-based radars and ice cores. While these are interesting studies, neither provides an estimate for the total annual GrIS snowfall accumulation, which is what we are referring to in this statement. To clarify, we have updated the sentence to say "wide range of estimates for total GrIS snowfall"

*xvi.* *26 "ground-based snow observations" io "ground-based observations"*
(Authors' Response) We have made this change.

*xvii.* *26 "can be useful to examine" io "are needed to look at"*
(Authors' Response) Modified to say "satellites are useful tools for looking at…".

*xviii.* *27 "from space"... "remotely" o*
(Authors' Response) In this case, the method/reference we mention is space-based, and it would not be appropriate to say "remotely", which could include many possible platforms (ship, aircraft, ground-based, etc.).

*xix.* *31 "an attractive" io "currently the best "*
(Authors' Response) We have modified the sentence to remove "currently the best", it now reads: "Satellite-borne active sensors are an advantageous platform for measuring the annual cycle of snowfall over the full GrIS because they can provide both information on falling snow as well as insight into the coincident clouds."

xx. *6 19 "explore the contents of clouds" rephrase to be more explicit re: "contents" "examine" may be better than "explore" ]*

(Authors' Response) We have modified this paragraph to be more explicit, and in the process, have removed an additional acronym (IWP) as it was no longer needed. It now reads: "In this work we use column-integrated reflectivity ($Z_{path}$ , $mm^6$ $m^{-2}$ ) as a proxy for the ice mass characteristics of the cloud. $Z_{path}$ is a relatively simple measurement related to the amount of hydrometeor backscatter (defined as $Z_{int}$ in Kulie et al., 2010; Pettersen et al., 2016).

xxi. *8 15 and throughout the paper, I suggest replacing "look" with "examine" ..."examined" io "looked specifically at"*

(Authors' Response) We have made this change.

xxii. *10 30 remove "(~1.2 million)" un-needed and arbitrary*

(Authors' Response) We have made this change.

xxiii. *9 16 "The frequency of detected snowfall events" io "The frequency of snowfall" 9*

(Authors' Response) We have made this change.

xxiv. *22-23 "more CLW cases along the western side of the GIS than the eastern" a point may be worth emphasizing as a new result this study brings forward. The previous information in the paragraph is already common knowledge in the field*

(Authors' Response) Thank you for the comment, we have included this as a point in the conclusions.

xxv. *9 25 winter "stronger north-south gradient compared to the annual distribution" also deserving some highlight, perhaps buried here, put it in conclusions if not also in abstract*

(Authors' Response) We have included this as a point in the conclusions.

xxvi. *10 5 "~83 % of its annual incoming solar insolation." according to what info?*

(Authors' Response) This was calculated from the 2B-FLXHR-lidar data product. We have added this to the text and included the relevant citation.

xxvii. *10 8 not in full agreement with "Snowfall accumulation is the only significant, positive term in the surface mass balance of the GIS" see major comments*

(Authors' Response) Addressed in major comments.

xxviii. *10 34 "hydrometeor" mean rain? be clear or use less jargon*

(Authors' Response) In this instance we were referring to ice particles, so have replaced "hydrometeor formation and/or growth" with "ice particle formation and/or growth".

xxix. *11 1-2 "while the common summer events are snowing at slightly higher rates on average, it is in winter that the less frequent, highest-intensity snowfall occurs." interesting point worth emphasizing in conclusions*

(Authors' Response) We have included this as a point in the conclusions.

xxx.   *11 13 "determine" io "see", modify throughout*
(Authors' Response) We have made this change.

xxxi.  *13 31-32 "Excluding a minor moisture recycling in the surface boundary layer that delivers frost during net surface radiative cooling events, the moisture required to produce GIS snowfall is not produced locally" io "The moisture required to produce GIS snowfall is not produced locally" ... actually the first sentence in 4.3 can be removed, the following sentence makes the point and there you might as well mention temperature inversion and PBD moisture recycling*
(Authors' Response) We have done as you suggest and removed the first sentence of Section 4.3, and modified the following sentence's beginning to smooth the new transition: "While many local factors influence when and where snowfall occurs over the GrIS (topography, surface type, temperature, etc.)…"

Authors' Response to Anonymous Referee #2

*1. (Referee #2 Comment – minor revisions) There are only a couple of very minor comments that the authors could address*

i.   *Page 3 Line 14: "been" is repeated.*
(Authors' Response) We have made this change.

ii.  *Page 12 Line 15: "have" is repeated.*
(Authors' Response) We have made this change.

iii. *Page 12 Line 30-35: you are talking here about opposite skewness of the distribution between winter and summer, but I don't actually see a negative skew in summer (with the peak to the right and the tail to the left). I would actually still see a tail to the right for summer even if steeper. I'll leave to the authors if considering this very minor comment or not.*
(Authors' Response) We see what you mean. The difference we were trying to emphasize was the movement of the peak in the distribution (from between 2-4 km in winter to between 3-6 km in summer) while the range of the distribution remained much the same. In light or your comment, we changed the text of page 12 line 30-32 to read: "There is also a change in the skewness of the distribution, with a positive skew (peak to the left, tail to the right) in the winter and a little to no skew (peak centered in the range of measurements) in summer."

iv.  *Page 13 Line 1: reading "a distinct nature of the two regimes" makes me think that I should be able to separate the two regimes from the histograms in fig. 9 and this actually not the case since the two distributions are quite totally overlapping. I then understand that these plots tell the story of the IWP behavior for the two regimes, but I would try to describe it better not to lead the reader to a wrong conclusion (being able to*

*distinguish between the regimes thanks to the dBZpath).*

(Authors' Response) Thank you for the comment. You are correct that the annual and winter distributions of dB(Zpath) are largely overlapping for the two regimes. The 'distinct nature' that we were referring to was in the peak locations and the seasonal behavior – in summer (Fig. 9c) CLW maintains the same shape as winter, while the IC events lean to larger values. We have changed the line beginning on page 13 line 1 to be clearer: "The histograms of dB(Zpath) (Fig. 9) highlight distinct seasonal behavior for the two regimes."

---

## Author Response (AR2)

We would like to thank the anonymous referee for their time and effort in reviewing and providing thoughtful feedback on our manuscript.

The reviewer comments are in red and our responses are in black.

Authors' Response to Anonymous Referee #3

This study uses satellite observations from CloudSat and CALIPSO to investigate snowfall regimes across the Greenland Ice Sheet. The classification of snowfall events across the ice sheet into ice-phase and mixed-phase cloud processes is novel and there are some interesting findings that may be useful for the ice sheet surface mass balance community. However, the manuscript requires major revisions before it can be considered for acceptance. The main problem is that the manuscript is poorly structured which makes it difficult to understand what literature gap the study aims to fill and how the authors did it. In the methods section, there is text about topographical basins in the "Satellite Data" section and text about averaging snowfall observations in the "Reanalyses" section. To fully understand the methods, the reader has to scroll up and down to find the relevant sentences. The manuscript is also lacking key details and there are many instances where the writing style should be more precise. Finally, one of the major novelties of the study is to divide the snowfall events in topographical basins. But this ignores extreme spatial gradients in air temperature. I think a better version of this study would be to separate the ice sheet by elevational as well as topographical regions. More detail is provided in my specific comments below.

(Authors' Response) Thank you for the suggestion of including an elevation analysis. In response, we conducted additional analysis and have added three plots to the manuscript to show how snowfall frequency, conditional snowfall rate, cloud depth, and dB(Z$_{path}$) vary with elevation for various regions and the two seasons. This elevation information adds an additional novel component to the manuscript and gives a more detailed understanding of how the two snowfall regimes behave over the GrIS.

In Section 4.1 we have added the following two paragraphs (appearing consecutively in the revised manuscript starting on line P10L31) and two figures:

[revised manuscript text omitted]

P1L22-P2L4: The first few lines of the introduction are muddled. The authors state that the Greenland Ice Sheet is important for the Earth's energy budget but cite a paper that finds that Greenland's impact on the cryosphere radiative forcing has not changed since 1979. Only later in the paragraph do the authors state the importance of the ice sheet for global sea-level rise. I recommend that the authors remove the first few sentences and focus the first paragraph on describing Greenland's recent and alarming contributions to global sea levels (i.e. from P2L4 onwards).

(Authors' Response) Thank you for the comment, we have removed the lines you suggest from the introduction.

P2L11: Only humans "build", recommend changing to "adding" or similar.

(Authors' Response) We have changed to "adding mass to".

P2L27: What is meant by "looked", please be more specific.

(Authors' Response) We have removed "looked" and modified the sentence. It now reads as follows: "*Both airborne and ground-based radars have been used to detect internal reflecting horizons below the surface to provide historical accumulations over the GrIS (Miège et al., 2013; Lewis et al., 2017), but those values are limited to specific transects, suffer complications from melt events, and only apply on annual or longer timescales.*"

P2L31-33: The lack of observational constraints is just one of many difficulties that have models have when modeling snowfall across the Greenland Ice Sheet. A better justification for this study would develop this statement more, especially given the prevalence of models used for Greenland SMB studies.

(Authors' Response) Thank you for the comment, we have modified the sentence to more clearly describe the cited study and more directly motivate the analysis that we did. "*Even with identical forcing, regional models have been shown to produce a wide range of GrIS precipitation amounts (Vernon et al., 2013), further highlighting the need for better understanding of the connections between large scale atmospheric conditions and snowfall.*"

P2L34: Again satellites do not "look" at snowfall, please be more specific. I noticed Anonymous Referee #1 also raised this point.

(Authors' Response) We have replaced "*looking at*" with "*detecting and quantifying*".

P3L1: This sentence does not make sense, how can "variations in surface emission" introduce errors into passive microwave estimates?

(Authors' Response) As described in the Liu and Curry (1997) paper we cite, remote sensing retrievals of snowfall from the passive satellite microwave data are based on upwelling microwave emission from the surface being diminished by scattering by ice and snow particles. These retrievals must assume a priori the upwelling surface emission, and so uncertainties in that emission contribute directly to uncertainties in the retrieval. Microwave emissivity varies substantially depending on surface conditions (fresh snow, old snow, ice, meltwater content) and it is these unknown conditions that cause uncertainties in the retrieved snowfall. From Liu and Curry (1997) regarding snowfall over sea ice, which is applicable to the GrIS as well: "…Precipitation over sea ice is also very valuable to climatological research. Satellite retrieval of precipitation may be possible over uniformly distributed multiyear ice. However, snow cover over the ice sheet would complicate the retrieval problem, because the scattering signal of falling snow is similar to that of the snow cover. Satellite retrieval of precipitation over younger ice is even more difficult due to the ice horizontal inhomogeneity."

Based on your comment we have expanded that sentence to be more clear about the methods. It now reads as follows: *"Surface snowfall can be estimated from space based on the upwelling microwave emission from the surface being diminished by scattering due to ice and snow particles in the atmosphere (Liu and Curry 1997). However, such satellite retrievals must assume a priori the upwelling surface emission, and since microwave emissivity varies substantially depending on surface conditions (fresh snow, old snow, ice, meltwater content) large uncertainties are introduced when the retrieval is applied over non-ocean surfaces."*

P3L5-7: The authors state that CloudSat has an advantage because it provides information on snowfall and precipitating clouds. But the previous paragraphs provided no information about why the latter should be important. The authors should consider moving P3L13-14 upwards to make this point.

(Authors' Response) Thank you for the note, we have moved lines P3L13 upwards two paragraphs.

P5L20-21: Would be useful to clarify the study period here? 2006 to 2016?

(Authors' Response) We have added the following to the sentence you indicate: "*which includes data between January 2007 and August 2016.*"

P5L21-22: 0.94 x 1.25 seems a strange choice of grid spacing, please justify. Why not a more conventional 1x1 degree grid spacing?

(Authors' Response) This spacing is the standard grid used by the Community Earth System Model (CESM). The dataset analyzed here was initially used in McIlhattan et al. (2017) to evaluate Arctic cloud and precipitation characteristics in the model. Though we do not compare results in this paper to CESM, we believe that this is an area of possible future work and this dataset will be useful to others that wish to compare to CESM. The analysis/conclusions presented here are not sensitive to small changes in the grid resolution so we do not feel that

re-binning the data to a 1x1 degree grid justifies the additional computation costs.

P5L23-31: This paragraph appears to be about the satellite products and drainage basins that are used to analyze the data which makes it confusing to read. Please separate.
(Authors' Response) We have moved the discussion of the drainage basins to the end of the section for clarity.

P6L13-14: What is meant by the "bin above the lowest for all profiles"? Please clarify.
(Authors' Response) The text has been modified to read as follows: "*In this study, when 2CSP identifies a satellite footprint as having potential contamination in the lowest bin that should be clutter free, we take the snowfall rate from the level immediately above, consistent with the methods of Palerme et al. (2019) and Milani et al. (2018).*"

P7L17: Does CloudSat/CALIPSO pass directly over Summit? If not, at what distance were CloudSat/CALIPSO data considered representative of Summit?
(Authors' Response) In this section (2.3 Ground-based Data), we are talking only about the surface based data. In the line you highlight, we are describing our method for mimicking the CloudSat/CALIPSO footprint by averaging the surface data. In Section 2.3 only data observed directly above Summit by the ground-based instrumentation is used.

P7L20: Why did you choose 5 m/s? Is it the average wind speed at Summit?
(Authors' Response) In the analysis of snowfall at Summit done by Pettersen et al. (2018), the authors found that the wind speeds during snowfall events were stronger than non-precipitating periods, and the majority of the time stronger than 5 m/s. We found in our analysis that that slower wind speeds resulted in more missed cases, so we chose to use 5 m/s for the averaging to be conservative in our estimate of detected cases.

In response to your comment, we have added the following text to the manuscript: "*Most wind speeds detected during Summit snowfall events are faster than 5 m s$^{-1}$ (P18), however, using faster wind speed thresholds (shorter time averaging) results in more detected cases so this slower threshold provides a conservative estimate of detection errors.*"

P8L1-3: Why is this text in the section titled "Reanalyses"? Please consider re-structuring this methods section.
(Authors' Response) Thank you for the comment. The way we use the reanalysis data involves choosing cases based on satellite information. For clarity, we have removed the information on how cases were selected from Section 2.4 and placed it in the results section where it will be more useful to readers.

P8L4: What is meant by the "nearest hourly ERA5 output"? Do the authors just take one grid cell from ERA5 for each basin or do you average across the basin?
(Authors' Response) We take the ERA output for the entire region, selecting the hourly data closest to the timing of a given snowfall event. To improve clarity, we have rewritten those lines to read as follows: "*In Section 4.3, we plot maps of the mean and anomaly of the 500 mb*

*geopotential height (GPH) and winds associated with snowfall events for the two regimes. The means are composites of the entire region using hourly ERA5 data nearest in time to the selected snowfall events. Climatological anomalies are generated by subtracting the long-term ERA5 monthly mean (1979-2018) from the hourly ERA5 data nearest in time to the selected snowfall events.*"

P8L8: What is meant by "ice habit"?
(Authors' Response) Habit refers to the various shapes that ice particles can take (hexagonal columns, 4-bullet rosette, dendrite snowflake, etc). We have modified the manuscript text to "*ice particle shape*" for clarity.

P10L10-11: Please define "summer months"? May- September?
(Authors' Response) "Summer months" in this work consists of May-September. We have revised the first sentence of this paragraph to clearly state this: "*In the summer months (defined here as May through September, consistent with P18…*"

P10L18-20: How do these figures compare to other studies?
(Authors' Response) We have added the following statement immediately after those lines to compare the distributions to other studies: "*The total annual snowfall in Fig. 5a is consistent with the results from Bennartz et al. (2019), with the highest fraction of GrIS snowfall occurring in basins 8.1 and 6.2. respectively. The largest accumulation in winter occurs in the basins along the southeastern coastline (Fig 5d), in agreement with other studies highlighting more and/or stronger cyclones (Zhang et al., 2004) and precipitation (e.g. Vihma et al., 2016; Berdahl et al., 2018) in that region in winter. In summer the largest accumulation is in the basins of the central west (Fig. 5g), a combination of both increased snowfall rate in the region and the relatively large area of these basins.*"

The total accumulation value listed in the figure is already compared in detail to other studies in the next paragraph.

P10L20-21: Defining two time periods of different length is very confusing for presenting the results. Suggest using the more conventional seasonal time periods.
(Authors' Response) Thank you for the comment. Our time periods were chosen to be consistent with the season definitions in Pettersen et al., 2018, the study which provided the motivation for this manuscript. The unusual season definitions relate to the GrIS meteorological annual cycle not conforming to the traditional definitions of the four seasons.

The summer (May – September) and winter (October – April) have been shown in previous work to have distinct atmospheric conditions in the central GrIS (Shupe et al., 2013; Castellani et al., 2015; Pettersen et al., 2018). The "summer-like" conditions (increased snowfall frequency, higher precipitable water vapor, more frequent liquid containing clouds, warmer temperatures, lower wind speeds, etc.) all persist through September. This season difference extends outside of the central GrIS as well. Zhang et al. (2004) used a similar two season definition, finding that winter (October – March) has greater cyclone intensity in the North

Atlantic and more storms close to the SE coast of Greenland relative to the summer (April - September),

Based on your comment, we have added the following paragraph to the beginning of our methods section: "*Throughout our analysis, we divide our data into two seasons: summer (May - September) and winter (October - April). This is consistent with the seasonal breakdown of P18 which was based on the distinct atmospheric conditions that occur at Summit during those time periods. Summer conditions in the central GrIS (increased snowfall frequency, higher precipitable water vapor, more frequent liquid containing clouds, warmer temperatures, lower wind speeds, etc.) all persist through September (Shupe et al., 2013; Castellani et al., 2015). This summer/winter distinction extends outside of the central GrIS as well. Zhang et al. (2004) used a similar two season definition (winter: October – March, summer: April - September), finding that winter has greater cyclone intensity in the North Atlantic and more storms close to the SE coast of Greenland relative to the summer.*"

P16L30: The annual accumulation value (399 Gt yr-1) identified in this study is associated with some serious bias due to missing snowfall events. The authors should state this here or remove this figure from the introduction because it does not represent a realistic estimate of Greenland snowfall.

(Authors' Response) All estimates of GrIS snowfall are subject to bias, no observing platform or model is yet perfectly able to capture precipitation characteristics over the full GrIS. The range in model and reanalysis estimates presented in Cullather et al. (2014) is ~581 – 899 Gt yr$^{-1}$ for the period 1980-2008 (calculated from the mean precipitation rates in Table 1 and GrIS area from Zwally et al. 2012), with ERA-Interim having a value at the lower end of that range (630 Gt yr$^{-1}$). There is evidence that reanalyses greatly overestimate southern coastal (Bromwich et al. 2015) and inland (Koyama and Stroeve 2019) GrIS snowfall relative to surface observations. Bromwich et al. (2015) Figure 6 shows ERA-Interim with an annual mean precipitation bias of ~50% for two stations on the southern coastline of Greenland, while Koyama and Stroeve (2019) show in their Table 3 that the Arctic System Reanalysis (ASR) has snowfall rates at Summit more than double that of the surface observations. Further, the analysis by Berdahl et al. (2018) found that the commonly used MAR regional climate model produces a mean high bias of 127 mm w.e. for the 1958-2012 period relative to surface stations in the southeastern GIS.

We recognize that our estimate is much lower than previous estimates, however there is much regional evidence that previous estimates have been biased high. While we do not argue that our estimate is "truth", we believe that it is a realistic estimate based on the only large-scale observations available, and it is worthwhile for our estimate to be part of the literature. We go to great lengths to describe the biases and limitations specific to our satellite data (Section 3), and don't believe there is a valid reason to remove our estimate from the introduction.

In response to your comment, we have added the following to the introduction (P2L25): "*There is evidence that reanalyses greatly overestimate southern coastal (Bromwich et al. 2016) and inland GrIS snowfall (Koyama and Stroeve, 2019) relative to available surface observations.*

*Bromwich et al. (2016) found the ERA-Interim reanalysis has an annual mean precipitation bias of ~50% for two stations on the southern coastline of Greenland, while Koyama and Stroeve (2019) found that the Arctic System Reanalysis has snowfall rates more than double that of the surface observations."*

…and the following to the section 3 (P9P30): *"However, it is not clear that the missed cases identified here should result in large scale biases in snowfall frequency or total accumulation values. Maahn et al. (2014) compared snowfall values derived from CloudSat and derived from ground based radar at sites in Norway and Antarctica, finding that the competing effects of shallow snowfall not being seen by the CPR and virga that was flagged as snowfall by the CPR though did not reach the surface resulted in CPR derived frequency being different by $\pm 5$ % and the total amount being underestimated by 9 - 11 %, relative to ground based values."*

…and the following to the conclusion (P19L16): *"We recognize that our total GrIS snowfall estimate of 399 Gt y$^{-1}$ is much lower than previously published estimates (e.g. Cullather et al. 2014), however there is evidence that GrIS snowfall has been overestimated relative to surface observations in both reanalyses (Bromwich et al., 2015; Koyoma and Stroeve, 2019) and models (Berdahl et al., 2018)."*

Figure 1: More information required. What does the numbering refer to? I presume the star is the location of Summit? Please clarify in the figure.
(Authors' Response) Thank you for bringing this to our attention, we have modified the figure caption to include that information. It now reads as follows: *"A summary of the CloudSat/CALIPSO satellite observations collected over the Greenland Ice Sheet (GrIS). The GrIS is divided into drainage basins as defined by the Ice Altimetry group at Goddard Space Flight Center (Zwally et al., 2012), the numbering of the basins corresponds to their numbering system. The color scale represents the total number of satellite overpasses in each basin during the full study period, January 2007 through August 2016. During that period, there were 14,703,887 individual satellite observations, 2,438,817 of which contained snowfall. The star indicates the location of Summit Station, and the circle is the 100 km radius surrounding Summit."*

Figure 2: Again, I do not understand why summer and winter represent different time spans. Either clarify this oddity or change the time periods to represent six months.
(Authors' Response) Please see the above discussion on why the time spans were chosen and the manuscript modifications made.

Figure 3: The divergent color-scale for tor panels A, D and G does not work well. Suggest using a sequential color-scale.
(Authors' Response) Thank you for the comment, we agree the color-scale in those panels was not ideal. We have updated to a more easily interpreted color-scale ('viridis'). We updated Figures 1 and 2 with this color-scale as well, and discretized the regime percent color bar in Figure 2 to be consistent with Figure 3.

[revised manuscript text omitted]

---

## Author Response (AR3)

We would like to thank the anonymous referee for their time and effort in reviewing our manuscript for the second time and providing additional feedback.

The reviewer comments are in red and our responses are in black.

Authors' Response to Anonymous Referee #3

This is my second review of "Satellite Observations of Snowfall Regimes over the Greenland Ice Sheet" by McIlhatten et al. The authors have generally done a good job of responding to my previous comments. The writing is more precise, the methods are mostly well-structured, and the division by elevation adds some additional insight. On my second reading I did still find a few minor things which could be improved. Note that my line numbers refer to the tracked changes version.

Specific comments

P1 L11: Suggest adding a "We find" to start this sentence so the transition to the results is clear (Authors' Response) We have made that change.

P1 L19-21: The abstract finishes with one of the least interesting results of the manuscript. Suggest removing and adding a concluding sentence that brings all these findings together. Perhaps something about implications for modeling Greenland Ice Sheet mass balance (which is what this study claims to do in the third sentence) or future behavior of the ice sheet. (Authors' Response) Thank you for the suggestion, we have removed the sentence you reference and replaced it with the following concluding sentence: "*When combined with future climate predictions, this snapshot of GrIS snowfall characteristics may shed light on how this source of ice sheet mass might respond to changing synoptic patterns in a warming climate.*"

P2 L4-18: Good scientific writing usually has some sort of order (from most to least important would work in this case). But in this introduction the importance of snowfall for accumulation is mentioned almost as an afterthought. This would be very confusing for readers with less expertise in this field. I'd argue that the role of snowfall for adding mass to the ice sheet is THE most important role in Greenland surface mass balance. Consider re-ordering. Having said that, I commend the authors for improving the rest of the introduction which now reads much better with the added text. (Authors' Response) Thank you for the suggestion, we have re-ordered as you suggest. The second paragraph of the introduction now reads as follows: "*Snowfall is responsible for both adding mass to and brightening the surface of the GrIS. While vapor deposition can be locally important, snowfall accumulation is by far the largest source term for the mass of the GrIS (Ettema et al., 2009; Bring et al., 2016). Snowfall rate, duration, and frequency of events are all important for accumulation. Surface brightness, or albedo, is largely dependent on the frequency of precipitation because fresh snow is more reflective than old snow and bare ice in the shortwave solar wavelengths (Petty, 2006; Box et al., 2012; Ryan et al., 2019). However, the shortwave albedo only matters during sunlit periods when there is incoming solar radiation;*"

*therefore the seasonal timing of snowfall events must also be considered. Fresh snowfall in summer can reduce absorbed shortwave by up to a factor of ~3, largely reducing local melt and meltwater runoff (Noël et al., 2015)."*

P2 L12: Suggest adding "and bare ice" after "old snow". Also the Enderlin et al. (2014) paper does not appear to have investigated drivers of albedo on the Greenland Ice Sheet. A better reference would be: Ryan et al. (2019). Greenland Ice Sheet surface melt amplified by snowline migration and bare ice exposure. Science Advances.
(Authors' Response) We have incorporated both the wording change and the Ryan et al. (2019) reference.

P2 L33-35: Suggest adding the following recent reference which also identified modeled snowfall bias over Greenland: Ryan et al. (2020). Evaluation of CloudSat's Cloud-Profiling Radar for Mapping Snowfall Rates Across the Greenland Ice Sheet. Journal of Geophysical Research.
(Authors' Response) Thank you for providing this reference, we have included it.

P6 L10-11: The methods are generally better structured now but this sentence seems a little out of place. I would add it next to the sentence P7 18-21.
(Authors' Response) We have moved the sentence to the location you suggest.

P7 L11-12: Suggest adding a sentence that outlines how 2BCCL retrieves cloud phase
(Authors' Response) We have added the following sentence: *"The relative strengths of the radar and lidar backscatters are distinct for each cloud phase: ice clouds produce weak to moderate lidar and strong radar backscatter; liquid clouds show strong lidar and weak radar backscatter; and the backscatter for both instruments is strong for mixed-phase clouds."*

P9 L7-8: Again this first sentence doe not fit with the rest of the paragraph which is actually about CloudSat's CPR. Then the next paragraph is about ICECAPS. Suggest removing mention of CloudSat in this first paragraph.
(Authors' Response) Thank you for pointing out that the beginning of this section is muddled. We have re-written it for clarity as follows:

*"In this section we investigate the reliability of CloudSat's CPR in detecting GrIS snowfall. Radar-derived snowfall rate estimates are dependent on assumptions about properties including ice particle shape, properties that are variable in space and time (Kulie et al., 2010). While the 2CSP snowfall rate is impacted by these assumptions, snowfall detection and resulting frequencies are independent of them. If the CPR detects precipitation size particles immediately above the blind zone, it is a good indicator that snow is falling at the surface (Boening et al., 2012; Milani et al., 2018; Palerme et al., 2019; Bennartz et al., 2019).*

*The ICECAPS instrument suite at Summit provides a unique opportunity to look at IC and CLW cases from below, which can provide insight into what is seen from above. When comparing 200 m and 700 m AGL snowfall reflectivities from an ICECAPS radar, Castellani et al. (2015) found evidence of growth..."*

P10 L19-23: Please also add discussion about how these results compare to Ryan et al. (2020). "Evaluation of CloudSat's Cloud-Profiling Radar for Mapping Snowfall Rates Across the Greenland Ice Sheet" who did a similar comparison with almost identical results.

(Authors' Response) Thank you for the additional reference, we have added the following sentence in that location: "*Similarly, Ryan et al., (2020) found that CPR derived snowfall rates correlate well with precipitation gauges at two locations on the surface of Greenland, and with CPR rates for two particular snowfall events coming within +/- 9 % of the precipitation gauge value.*"

P19 L23: Unless I am missing something, the word "conditional" is meaningless here. Consider removing.

(Authors' Response) The "conditional" highlights that we are not including non-precipitating observations in the mean rate. However, since we are clear in the Fig. 4 caption how we obtained the rates, we have removed the word here to avoid confusion.

P28: "the numbering of the basins corresponds to their numbering system" is unnecessary because already implied.

(Authors' Response) We have removed that phrase.

P31: Ah so this what conditional refers to. Why don't you just say, "mean snowfall rate for given regime"? If you are going to stick with "conditional" please also clarify in P7 L7-13.

(Authors' Response) We will keep conditional for the y-axis in Fig. 4, since it is described in the figure caption and we want to be clear that we are not including non-precipitating observations in the mean. The conditional snowfall rates in Fig. 4 are described in the results section on P12 L7-13 (P7 L7-13 describe one of the satellite products), so we presume that is where you would like us to add clarification.

To avoid confusion, we have removed "conditional" from the results text, which now reads as follows: "*
[revised manuscript text omitted]